# Inhibition of microtubule detyrosination by parthenolide facilitates functional CNS axon regeneration

Marco Leibinger[1,2†], Charlotte Zeitler[1,2†], Miriam Paulat[2], Philipp Gobrecht[1,2], Alexander Hilla[2], Anastasia Andreadaki[1,2], Rainer Guthoff[3], Dietmar Fischer[1,2]*

[1]Center for Pharmacology, Institute II, Medical Faculty and University of Cologne, Cologne, Germany; [2]Department of Cell Physiology, Ruhr University of Bochum, Bochum, Germany; [3]Eye Hospital, Heinrich Heine University Düsseldorf, Düsseldorf, Germany

*For correspondence:
dietmar.fischer@uni-koeln.de

†These authors contributed equally to this work

Competing interest: The authors declare that no competing interests exist.

**Abstract** Injured axons in the central nervous system (CNS) usually fail to regenerate, causing permanent disabilities. However, the knockdown of *Pten* knockout or treatment of neurons with hyper-IL-6 (hIL-6) transforms neurons into a regenerative state, allowing them to regenerate axons in the injured optic nerve and spinal cord. Transneuronal delivery of hIL-6 to the injured brain stem neurons enables functional recovery after severe spinal cord injury. Here we demonstrate that the beneficial hIL-6 and *Pten* knockout effects on axon growth are limited by the induction of tubulin detyrosination in axonal growth cones. Hence, cotreatment with parthenolide, a compound blocking microtubule detyrosination, synergistically accelerates neurite growth of cultured murine CNS neurons and primary RGCs isolated from adult human eyes. Systemic application of the prodrug dimethylamino-parthenolide (DMAPT) facilitates axon regeneration in the injured optic nerve and spinal cord. Moreover, combinatorial treatment further improves hIL-6-induced axon regeneration and locomotor recovery after severe SCI. Thus, DMAPT facilitates functional CNS regeneration and reduces the limiting effects of pro-regenerative treatments, making it a promising drug candidate for treating CNS injuries.

## eLife assessment

The primary goal of this paper is to examine microtubule detyrosination as a potential therapeutic target for axon regeneration. The **valuable** findings of this study provide **convincing** evidence for mechanistic links between microtubule detyrosination and neurite outgrowth in vitro and some evidence for axon regeneration in vivo.

## Introduction

Traumatic CNS injuries often lead to irreversible impairments, such as blindness or loss of motor and sensory function, due to the inability of severed axons to regrow. This regenerative failure is mainly caused by an insufficient intrinsic growth capacity of mature CNS neurons and the inhibitory environment for axonal growth cones at the injury site (*Silver and Miller, 2004*; *Fischer and Leibinger, 2012*; *Lu et al., 2014*).

Although no clinically applicable treatment is available to human patients, various strategies exist to promote axon regeneration in experimental models of CNS injury (*Silver and Miller, 2004*; *Fischer and Leibinger, 2012*; *Zheng and Tuszynski, 2023*). For example, targeting the inhibitory environment at the lesion site by neutralizing myelin-derived factors or chondroitin sulfate proteoglycans

(CSPGs) led to axonal regeneration and, in some cases, induced functional recovery after incomplete spinal cord lesions (*Moon et al., 2001*; *Bradbury et al., 2002*; *Cafferty and Strittmatter, 2006*; *Lee et al., 2010*). On the other hand, it became evident that not only disinhibition but also strategies to stimulate cell-intrinsic signaling pathways are essential to promote robust axon growth. Among these, lens injury-induced expression and release of the IL-6-type cytokines ciliary neurotrophic factor (CNTF) and leukemia inhibitory factor (LIF) from retinal astrocytes transforms retinal ganglion cells (RGCs) into a regenerative state and enables moderate axon growth in the injured optic nerve (*Fischer et al., 2000*; *Yin et al., 2003*; *Leibinger et al., 2009*; *Müller et al., 2009*). The designer cytokine hyper-IL-6 (hIL-6), a fusion protein of IL-6 and the soluble IL-6 receptor α (IL-6Rα) can enormously increase these effects. In contrast to natural cytokines, hIL-6 directly binds to the gp130 receptor and does not rely on previous interaction with cytokine-specific alpha-receptors (*Fischer et al., 1997*; *Heinrich et al., 2003*; *Leibinger et al., 2016*; *Terheyden-Keighley et al., 2022*). This enables a more potent activation of downstream signaling pathways, such as the Janus kinase/signal transducer and activator of transcription 3 (JAK/STAT3) and phosphoinositide 3-kinase/protein kinase B (PI3K/AKT) pathways (*Leibinger et al., 2016*). As a result, AAV-mediated hIL-6 expression in RGCs elicits a more vigorous optic nerve regeneration over long distances than most other single treatments (*Leibinger et al., 2016*; *Fischer, 2017*). Similarly, hIL-6 expression in corticospinal neurons simultaneously promotes axon regeneration of multiple descending tracts and partially restores motor function after a complete spinal cord crush (*Leibinger et al., 2021*; *Terheyden-Keighley et al., 2022*).

Another way to stimulate CNS axon regeneration is to directly activate the PI3K/AKT pathway by knockout of phosphatase and tensin homolog ($Pten^{-/-}$) (*Park et al., 2008*; *Liu et al., 2010*; *Zukor et al., 2013*). This approach stimulated axon regeneration in the corticospinal tract even when applied one year after injury and induced functional recovery of voluntary skilled movements after incomplete spinal cord injury or pyramidotomy (*Liu et al., 2010*; *Lewandowski and Steward, 2014*; *Danilov and Steward, 2015*; *Du et al., 2015*). $Pten^{-/-}$ or hIL-6-mediated AKT activation causes phosphorylation of several substrates, including inactivating phosphorylation of glycogen synthase kinase 3 (GSK3) (*Sutherland et al., 1993*; *Sutherland and Cohen, 1994*). The specific GSK3β knockout ($Gsk3b^{-/-}$) in RGCs facilitates optic nerve regeneration by reducing collapsin response mediator protein 2 (CRMP2) phosphorylation in axons, while constitutively active GSK3 ($Gsk3^{S/A}$) compromises $Pten^{-/-}$ and cytokine-induced regeneration, underlining the relevance of the AKT/GSK3/CRMP2 pathway for CNS regeneration (*Leibinger et al., 2017*; *Leibinger et al., 2019*).

Besides CRMP2, GSK3 phosphorylates microtubule-associated protein 1B (MAP1B). Phosphorylated MAP1B keeps microtubules in a dynamic state by blocking the removal of the C-terminal tyrosine from α-tubulin subunits (detyrosination) in axonal growth cones (*Lucas et al., 1998*; *Goold et al., 1999*; *Gonzalez-Billault et al., 2001*; *Owen and Gordon-Weeks, 2003*; *Gobrecht et al., 2014*; *Gobrecht et al., 2016*). In addition, inhibition of tubulin detyrosination accelerates peripheral nerve regeneration (*Gobrecht et al., 2016*). This was achieved by local or systemic application of the sesquiterpene lactone parthenolide, an inhibitor of VASH1/2, which mediates detyrosination (*Fonrose et al., 2007*; *Aillaud et al., 2017*; *Nieuwenhuis et al., 2017*; *Gobrecht et al., 2022*).

The current study addressed whether parthenolide affects axon regeneration in the CNS. It shows that it markedly accelerates neurite outgrowth of primary CNS neurons by reducing tubulin detyrosination in axonal tips. Moreover, intraperitoneal application of its prodrug dimethylamino-parthenolide (DMAPT) facilitates axon regeneration after optic nerve and spinal cord injuries *in vivo*. Furthermore, hIL-6 and $Pten^{-/-}$ increased tubulin detyrosination. At the same time, parthenolide antagonized this effect, further enhancing hIL-6-induced optic nerve and spinal cord regeneration and functional recovery of hindlimb locomotion after severe spinal cord injury. The effects of parthenolide on detyrosination and neurite outgrowth were reproducible in human adult RGCs. In particular, parthenolide is an ideal cotreatment to accelerate axon regeneration and functional recovery in combination with other treatment strategies, making it a promising drug candidate for treating nerve injuries.

## Results

### Parthenolide and CNTF synergistically promote neurite growth of adult RGCs

In contrast to adult RGCs, sensory neurons usually show spontaneous neurite growth (*Müller et al., 2009*; *Leibinger et al., 2013*; *Leibinger et al., 2016*; *Levin et al., 2016*). Moreover, parthenolide-induced inhibition of tubulin detyrosination in axonal growth cones markedly accelerates the regeneration of sensory neurons in culture and *in vivo* (*Gobrecht et al., 2014*; *Gobrecht et al., 2016*). To investigate whether parthenolide has similar effects on CNS neurons, we cultured murine RGCs in the presence of increasing compound concentrations for 4 days. As in sensory neurons (*Gobrecht et al., 2016*), parthenolide significantly enhanced neurite extension concentration-dependently with maximum effects at 0.5 and 1 nM (*Figure 1A and B*). Expectedly RGC survival remained unaffected (*Figure 1C*) since RGCs do not undergo axotomy-induced cell death as early as 4 days after culturing. Moreover, cotreatment of parthenolide with CNTF showed a significant additive effect (*Figure 1D and E*), resulting in a 3.5-fold higher average neurite length than vehicle-treated controls (*Figure 1D and E*). Although parthenolide reportedly reduces JAK/STAT3 signaling at micromolar concentrations (*Sobota et al., 2000*; *Skoumal et al., 2011*), the nanomolar concentrations used in our experiments did not alter CNTF-induced STAT3 phosphorylation in RGCs (*Figure 1F and G*). Conversely, CNTF did not measurably affect parthenolide's inhibitory effect on microtubule detyrosination in axon tips (*Figure 1H,I*). We also tested parthenolide on human RGCs. To this end, we prepared and cultured primary RGCs from adult human eyes. Their identity as human RGCs was verified by retinal wholemount staining, showing βIII-tubulin-positive cells in the ganglion cell layer and axon bundles in the fiber layer, as seen in murine retinae (*Figure 1—figure supplement 1A*). Furthermore, their morphology was comparable to murine RGCs, while the cell diameter of human RGCs was larger as observed in cultures and in retinal wholemounts. (*Figure 1—figure supplement 1A,B*).Viability of cultured human RGCs was confirmed by their ability to express GAP43 and to grow neurites (*Figure 1—figure supplement 1B,C*). Parthenolide and CNTF significantly enhanced the average neurite length at similar concentrations as in mouse RGCs, while cotreatment enhanced neurite growth sixfold compared to controls after 4 days (*Figure 1J and K*). The total numbers of RGCs per well remained unaffected (*Figure 1L*). Parthenolide also reduced microtubule detyrosination in axonal growth cones (*Figure 1M and N*).

### Parthenolide counteracts hIL-6 and *Pten*[-/-]-induced microtubule detyrosination

Due to its direct binding of the signal-transducing gp130 receptor, hIL-6 promotes neurite growth of RGCs more efficaciously than CNTF and, contrary to CNTF, also measurably induces PI3K/AKT signaling in these neurons (*Leibinger et al., 2016*). However, AKT inactivates GSK3 (*Figure 2A*), which promotes CRMP2 activity (*Leibinger et al., 2017*; *Leibinger et al., 2019*). While enhanced CRMP2 activity is favorable for axon growth (*Liz et al., 2014*; *Leibinger et al., 2017*), GSK3 inhibition might also elevate tubulin detyrosination via reduced MAP1B activation (*Figure 2A*). However, as shown in the previous experiments (*Figure 1*), reducing microtubule detyrosination promotes neurite growth, suggesting that this effect of AKT/GSK3 might limit optimal axonal growth. We intravitreally injected AAV2-hIL-6 into mice 3 weeks before tissue isolation to test these possibilities. Subsequent western blot analysis of respective optic nerve lysates indeed revealed increased inhibitory phosphorylation of both GSK3 isoforms (pGSK3α, pGSK3β) by hIL-6 compared to untreated controls, without affecting total GSK3 expression (*Figure 2B–E*). Moreover, while hIL-6 reduced inhibitory phosphorylation of CRMP2 at threonine 514 (T514) without changing total CRMP2 levels (*Figure 2B, F and G*), it expectedly elevated tubulin detyrosination (*Figure 2B and H*). Since parthenolide specifically acts on tubulin detyrosination without influencing CRMP2 activity (*Figure 2—figure supplement 1A,B*), it should be suitable to counteract an hIL-6-mediated increase in axonal tubulin detyrosination. To test this hypothesis, we cultured RGCs from AAV-hIL-6-treated mice or respective controls for 4 days without or in the presence of different parthenolide concentrations. Afterward, we measured the percentage of axon tips with detyrosinated tubulin. Notably, DMAPT does not affect long-lived pre-existing microtubules because, within these, tubulin is present mainly in the detyrosinated form already (*Gundersen and Bulinski, 1986*; *Gundersen et al., 1987*; *Akhmanova and Maiato, 2017*). However, DMAPT

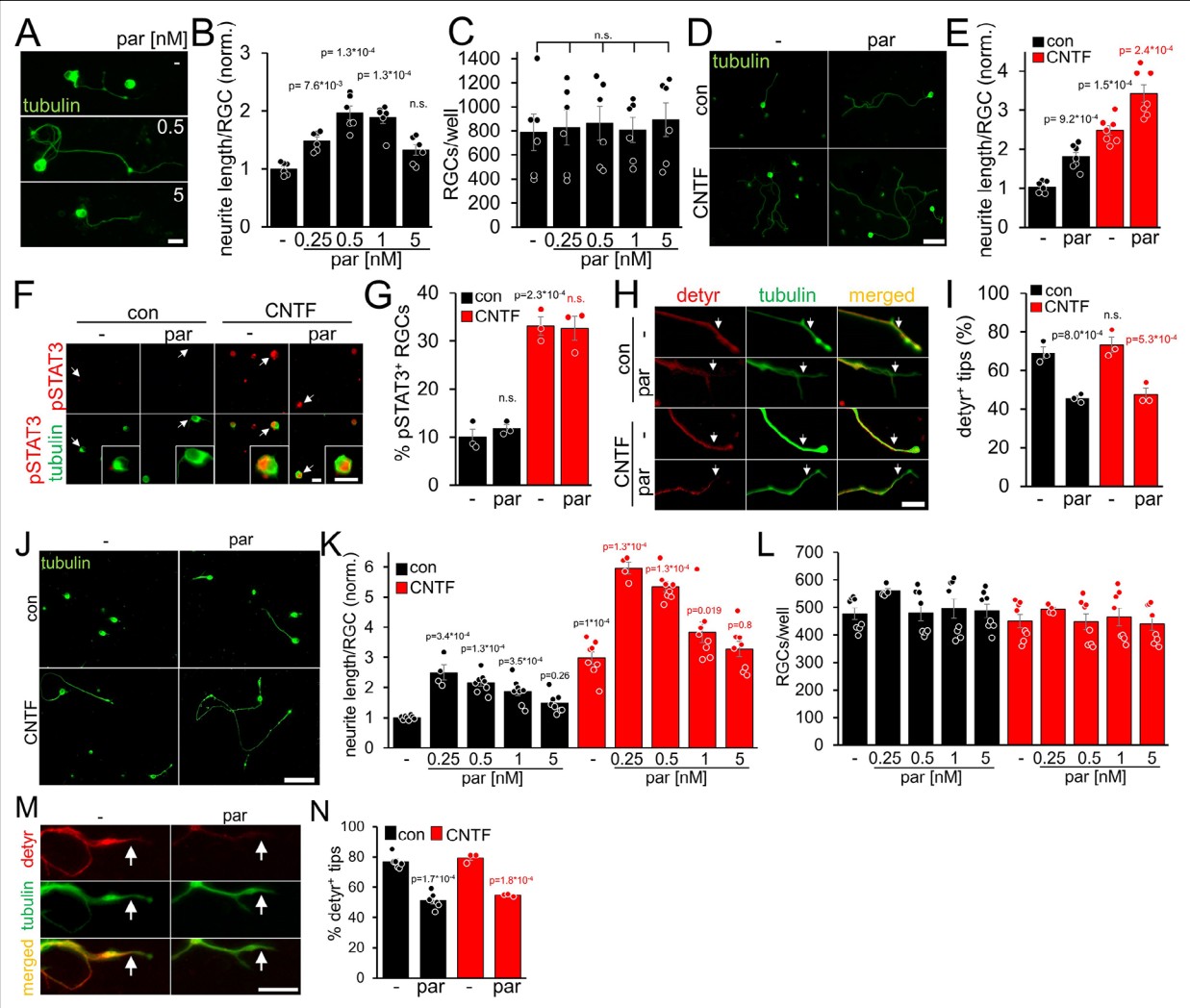

**Figure 1.** Parthenolide and CNTF synergistically promote neurite growth of murine and primary adult human RGCs. (**A**) Images of βIII-tubulin (tubulin) positive RGCs cultured for 4 days in the presence of either vehicle (-) or parthenolide (par; 0.5 nM or 5 nM). Scale bar: 15 μm (**B**) Quantification of neurite growth of cultures described in **A** with par concentrations between 0.25 and 5 nM. Data were normalized to untreated controls with an average neurite length of 4.6 μm per RGC and represent means ± SEM of six independent experiments. (**C**) Quantification of RGC numbers per well in cultures described in **A**. (**D**) Images of βIII-tubulin positive RGCs cultured for 4 days in the presence of vehicle (-) or par (0.5 nM), either without (con) or in combination with CNTF (200 ng/ml). Scale bar: 50 μm. (**E**) Quantification of neurite growth in cultures described in **D**. Data were normalized to untreated controls with an average neurite length of 9.6 μm per RGC and represent means ± SEM of seven independent experiments. (**F**) Representative images of βIII-tubulin (tubulin; green) positive RGCs and phosphorylated STAT3 (pSTAT3, red) in cultures described in **D**. Insets show higher magnifications of selected cells, indicated by white arrows. CNTF, but not par, induced STAT3 phosphorylation. Scale bars: 15 μm (**G**) Quantification of pSTAT3 positive RGCs in cultures described in **D**. Only CNTF, but not par, affected pSTAT3 levels. Data represent means ± SEM of three independent experiments. (**H**) Representative images of RGC neurites from cultures described in **D**, immunohistochemically stained for βIII-tubulin (green) and detyrosinated α-tubulin (detyr, red). White arrows indicate the last 15 μm from axon tips after respective treatments. Scale bar: 15 μm. (**I**) Percentages of detyrosinated tubulin-positive (detyr⁺) axon tips. Par but not CNTF reduced detyrosination. Data represent means ± SEM of three independent experiments. (**J**) βIII-tubulin (tubulin) stained human RGCs isolated from the eyes of adult human patients and cultured for 4 days in the presence or absence of CNTF (200 ng/ml) and/or parthenolide (par, 0.5 nM). Scale bar: 100 μm. (**K**) Quantification of the average neurite length per RGC in cultures depicted in **J**. Par was applied at indicated concentrations, either alone or combined with CNTF. Data were normalized to untreated controls with an average neurite length of 16.3 μm per RGC and represent means ± SEM of four technical replicates, each from two independent experiments with individual human eyes (n=8). (**L**) Quantification of RGCs per well in cultures described in **J**, **K**. (**M**) Images of axon tips from RGCs as described in **J**. Cells were cultured in the absence (-) or presence of par (0.5 nM) and then immunostained for detyrosinated α-tubulin (detyr, red) and βIII-tubulin (tubulin, green). White arrows indicate detyrosinated tubulin-positive or negative neurite tips in control and the par-treated groups. Scale bar: 10 μm. (**N**) Quantification of axon tips containing detyrosinated tubulin from RGCs described in **J**. Thirty neurite tips per group were analyzed in three technical replicates from one or two independent experiments with individual human eyes (n=3–6). Significances of intergroup differences in B, C, E, G, I, K, and N were analyzed using a one-way (**B**, **C**), or two-way (**E, G, I, K, N**) analysis of variance (ANOVA) followed by the Tukey or Holm-Sidak post hoc test. P-values indicate statistical significance

*Figure 1 continued on next page*

*Figure 1 continued*

compared to the untreated (black p-values) or the CNTF-treated (red p-values) groups. n.s.=non-significant. Dots in **B**, **C**, **E**, **G**, and **I** represent values from at least three independent experiments. Dots in **K**, **L**, and **N**, represent values from technical replicates, each from two independent experiments.

The online version of this article includes the following figure supplement(s) for figure 1:

**Figure supplement 1.** Verification of human RGCs.

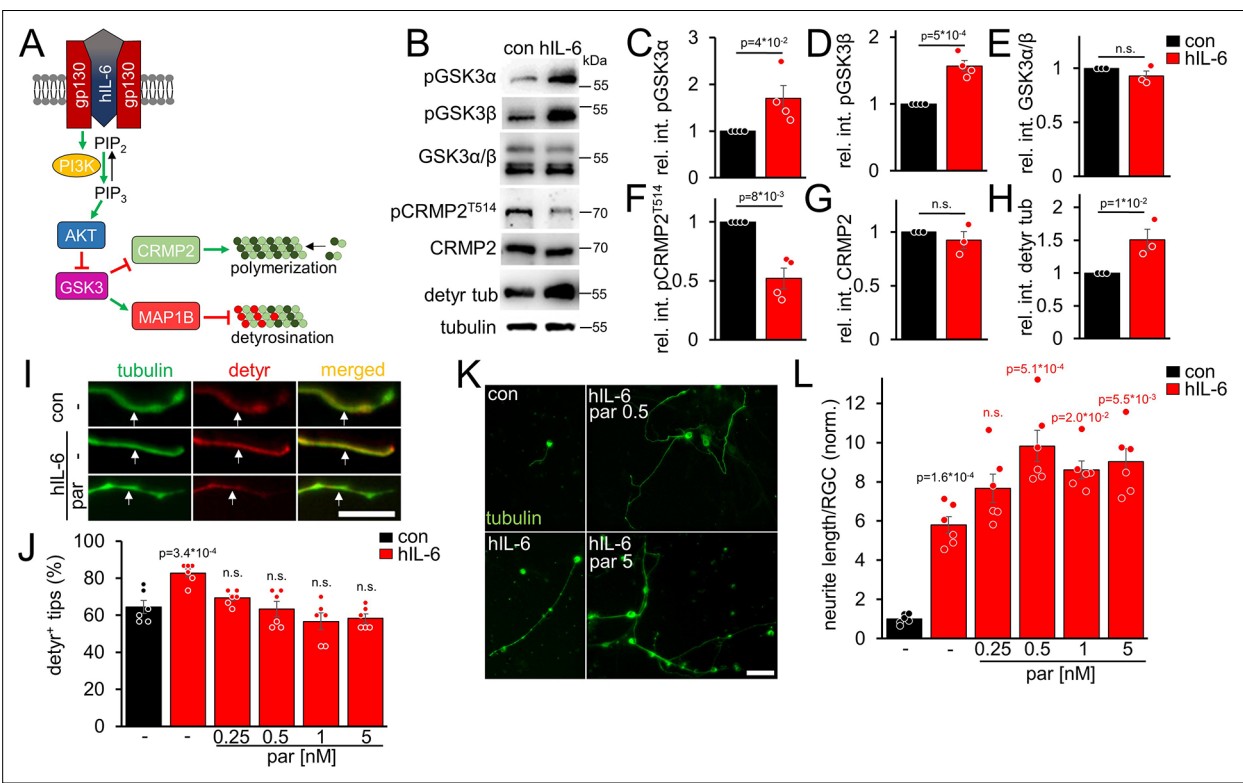

**Figure 2.** hIL-6 elevates microtubule detyrosination. (**A**) Schematic drawing illustrating the effect of hIL-6 on the PI3K/AKT/GSK3 signaling pathway. hIL-6 binding to gp130 activates phosphatidylinositol 3-kinase (PI3K), converting phosphatidylinositol (4, 5)-bisphosphate (PIP₂) to phosphatidylinositol (3, 4, 5)-trisphosphate (PIP₃). PIP₃ stimulates AKT. Subsequent effects on GSK3, CRMP2, and microtubule detyrosination are shown in **B–L**. (**B**) Western blots of optic nerve lysates from untreated mice (con) or animals that had received intravitreal AAV2-hIL-6 injections 3 weeks earlier. Retinal hIL-6 expression elevated inhibitory phosphorylation of GSK3α and GSK3β, while total GSK3 levels remained unaffected. Inhibitory CRMP2 phosphorylation was reduced without altering total CRMP2 levels, while detyrosinated tubulin levels were increased. βIII-tubulin (tubulin) served as a loading control. (**C–H**) Densitometric quantification of western blots shown in **B** relative to βIII-tubulin and normalized to the untreated control (con). Data represent means ± SEM of samples from at least three animals per group. (**I**) Representative images of axon tips from dissociated RGCs from mice, as described in **B**. RGCs, were cultured for 4 days in the presence of either vehicle (-) or parthenolide (par; 5 nM) and were immunohistochemically stained for βIII-tubulin (tubulin, green) and detyrosinated α-tubulin (detyr, red). White arrows indicate the last 15 μm of detyrosinated or negative axon tips after respective treatments. Scale bar: 15 μm (**J**) Quantification of the percentages of detyrosinated tubulin-positive (detyr⁺) axon tips. Par was applied at indicated concentrations. hIL-6 treatment increased tubulin detyrosination, while par blocked this effect. Data represent means ± SEM of 3 technical replicates from 2 independent experiments. P-values refer to untreated control. (**K**) Representative images of βIII-tubulin (tubulin) positive RGCs from cultures as described in **I**, cultured in the presence or absence of parthenolide (par; 0.5 nM or 5 nM). Scale bar: 50 μm (**L**) Quantification of the average neurite length per RGC in cultures depicted in **K**. Par was applied at indicated concentrations in combination with hIL-6. Data were normalized to untreated controls with an average neurite length of 8.4 μm per RGC and represent means ± SEM of three technical replicates from two independent experiments. Significances of intergroup differences were evaluated using Student's t-test for quantifications shown in **C-H**. A one-way analysis of variance (ANOVA) followed by a Holm-Sidak or Tukey post hoc test is shown in **J** and **L**, respectively. P-values indicate statistical significance compared to the untreated (black p-values) or the vehicle-treated hIL-6 (red p-values) groups. n.s.=non-significant. Dots represent values from single animals in **C-H** and three technical replicates of two independent experiments, respectively, in **J and L**.

The online version of this article includes the following figure supplement(s) for figure 2:

**Figure supplement 1.** Parthenolide does not affect CRMP2 phosphorylation, STAT3 phosphorylation, or mTOR signaling.

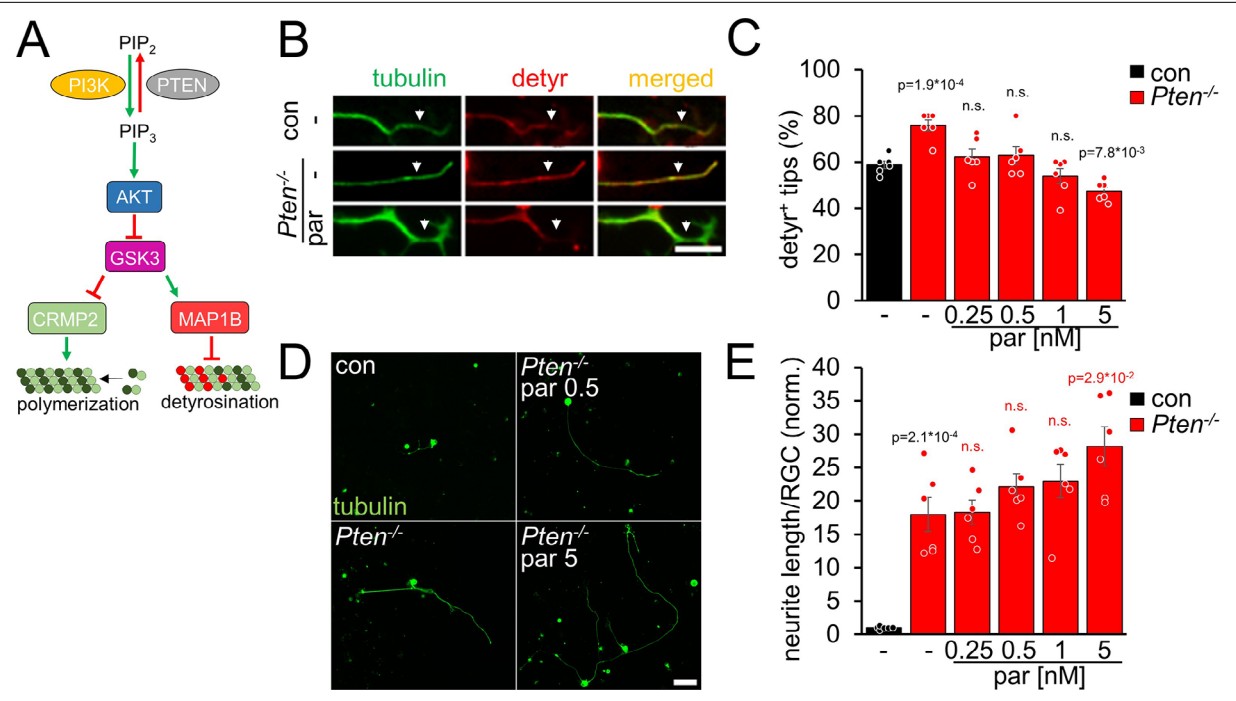

**Figure 3.** *Pten⁻/⁻* increases microtubule detyrosination. (**A**) Schematic drawing illustrating the effect of *Pten⁻/⁻* on the PI3K/AKT/GSK3 signaling pathway. *Pten⁻/⁻* promotes the conversion of phosphatidylinositol (4, 5)-bisphosphate (PIP₂) to phosphatidylinositol (3, 4, 5)-trisphosphate (PIP₃). PIP₃ stimulates AKT and thus induces inhibitory GSK3 phosphorylation. As a result, the phosphorylation of CRMP2 and MAP1B is reduced. The effect on tubulin detyrosination is illustrated in **B–E**. (**B**) Representative images of axon tips from dissociated RGCs of *Pten*-floxed mice. Animals were either untreated (con) or had received intravitreal injections of AAV2-Cre 3 weeks before dissociation (*Pten⁻/⁻*). RGCs were cultured for 4 days in the presence of either vehicle (-) or parthenolide (par; 5 nM) and were immunohistochemically stained for βIII-tubulin (tubulin, green) and detyrosinated α-tubulin (detyr, red). White arrows indicate the last 15 μm of axon tips. Scale bar: 15 μm (**C**) Quantification of the percentages of detyrosinated tubulin-positive (detyr ⁺) axon tips. Par was applied at indicated concentrations. *Pten⁻/⁻* increased tubulin detyrosination, while par blocked this effect. Data represent means ± SEM of 3 technical replicates from 2 independent experiments. (**D**) Representative images of βIII-tubulin (tubulin) positive RGCs from cultures as described in **B**, cultured in the presence or absence of parthenolide (par; 0.5 nM or 5 nM). Scale bar: 50 μm (**E**) Quantification of the average neurite length per RGC in cultures depicted in **D**. Par was applied at indicated concentrations in combination with *Pten⁻/⁻*. Data were normalized to untreated controls with an average neurite length of 1.04 μm per RGC and represent means ± SEM of three technical replicates from two independent experiments. The significance of intergroup differences was evaluated through a one-way analysis of variance (ANOVA) followed by a Holm-Sidak or Tukey post hoc test for data shown in **C and E**, respectively. P-values indicate statistical significance compared to the untreated (black p-values) or the vehicle-treated *Pten⁻/⁻* (red p-values) groups. n.s.=non-significant. Dots represent values from three technical replicates of two independent experiments.

The online version of this article includes the following figure supplement(s) for figure 3:

**Figure supplement 1.** Parthenolide does not compromise *Pten⁻/⁻*-mediated mTOR pathway activation.

can prevent the detyrosination of newly attached tubulin dimers in the growth cones of developing axons. Confirmingly, microtubule detyrosination in axonal tips was significantly higher in hIL-6-treated cultures than in controls, and parthenolide cotreatment reversed this effect (*Figure 2I and J*). Furthermore, the increased detyrosination in hIL-6-treated neurons also affected the effective parthenolide concentration on neurite extension in these cultures. Compared to the control cultures, where parthenolide showed the most substantial increase between 0.5 and 1 nM, followed by a decrease at 5 nM (*Figure 1A and B*), neurite growth of hIL-6-stimulated RGCs was increased even at higher parthenolide concentrations (up to 5 nM) (*Figure 2K and L*). The hIL-6-induced STAT3 phosphorylation remained unaffected by any parthenolide concentration applied (*Figure 2—figure supplement 1C,D*).

AKT is also activated by *Pten⁻/⁻*, another regeneration-promoting approach (*Park et al., 2008*; *Liu et al., 2010*; *Bei et al., 2016*; *Leibinger et al., 2019*). To investigate microtubule detyrosination in *Pten⁻/⁻* RGC neurites, we prepared RGC cultures from *Pten*-floxed mice after intravitreal AAV2-Cre injection. As expected, *Pten⁻/⁻* enhanced tubulin detyrosination in axon tips (*Figure 3A, B and C*). This effect was counteracted by parthenolide, which further improved neurite outgrowth of *Pten⁻/⁻*-RGCs concentration-dependently (*Figure 3B, C, D and E*). Moreover, activation of the mTOR

pathway, a significant contributor to the *Pten*[-/-]-mediated effects on axon growth, was not compromised by parthenolide, reflected by unaltered activation of ribosomal protein S6 in these RGC cultures (*Figure 3—figure supplement 1A,B*). Thus, parthenolide counteracts AKT/GSK3-induced microtubule detyrosination and synergistically enhances hIL-6 or *Pten*[-/-]-stimulated neurite growth.

## Dimethylamino-parthenolide accelerates optic nerve regeneration

To test whether parthenolide can accelerate CNS regeneration *in vivo*, we used its water-soluble prodrug dimethylamino-parthenolide (DMAPT), which can cross the blood-brain barrier (*Guzman et al., 2007*; *Neelakantan et al., 2009*; *Hexum et al., 2015*). To this end, adult mice were subjected to an optic nerve crush (ONC) and received daily intraperitoneal injections of either vehicle or DMAPT (2 µg/kg) (*Figure 4A*). The number of CTB-labeled, regenerated axons was quantified in optic nerve sections 2 weeks post-injury. Compared to vehicle-treated controls, more axons reached distances of up to 0.5 mm distal to the lesion site after DMAPT application, indicating a slight but significant effect on regeneration (*Figure 4B–E*). Expectedly, DMAPT was not able to protect RGCs from axotomy-induced cell death (*Figure 4F and G*) since it does solely accelerate microtubule polymerization in axonal growth cones without affecting neuroprotective signaling pathways in the cell body (*Figure 1F and G*; *Figure 2—figure supplement 1*). We then repeated these experiments in combination with intravitreally applied AAV2-hIL-6 which reportedly has a significant neuroprotective effect (*Leibinger et al., 2016*; *Figure 4H*). Two weeks after injury, retinal cross-sections revealed robust phosphorylation of STAT3 in AAV2-hIL-6-treated RGCs, which remained unaffected by DMAPT treatment (*Figure 4I*). As reported previously (*Leibinger et al., 2016*), hIL-6 treatment alone enabled substantial regeneration with CTB-labeled axons reaching distances up to 1.5 mm 2 weeks after injury (*Figure 4J–O*). Notably, additional DMAPT application at 2 µg/kg increased the number of axons at 1.5 mm fivefold, with some axons reaching distances of 2.5 mm within the same time after injury (*Figure 4J, K, N and O*). As observed in RGC cultures, DMAPT acted dose-dependently, with less pronounced effects at lower (0.2 µg/kg) or higher (20 µg/kg) dosages (*Figure 4—figure supplement 1A,B*). Thus, the systemic application of DMAPT sufficiently enhances and accelerates the effect of hIL-6 on optic nerve regeneration.

## Systemic DMAPT application accelerates raphespinal regeneration after spinal cord crush

Hyper-IL-6-mediated axonal regeneration of raphespinal fibers enables functional recovery after a severe spinal cord crush (SCC) (*Leibinger et al., 2021*). Therefore, we investigated whether systemic DMAPT treatment also affects raphespinal regeneration or functional recovery. To this end, we performed a complete SCC at the thoracic vertebra 8 (T8) level, followed by daily intraperitoneal injections of either vehicle or DMAPT at 2 µg/kg (*Figure 5A*). Hindlimb movement was analyzed using the Basso Mouse Scale (BMS) (*Basso et al., 2006*) over 8 weeks after injury. One day after SCC, BMS scores dropped to 0, indicating the absence of hindlimb movement (*Figure 5B and C*). After 2 weeks, most vehicle-treated animals recovered from the injury-induced spinal shock and reestablished typical extensive ankle movement (BMS: 2) (*Figure 5B and C*; *Video 1*). In contrast, DMAPT treatment slightly but significantly improved functional recovery. Most of these animals showed plantar hind paw placement as early as 2 weeks after injury, occasionally accompanied by bodyweight support (BMS: 3) (*Figure 5B and C*; *Video 2*). For this reason, the spinal cord tissue of some animals was analyzed for serotonergic fiber regeneration at this early time. The absence of serotonergic fibers at ~10 mm distal to the lesion site verified the lesion completeness in each animal included in this study to rule out the presence of spared raphespinal axons (*Figure 5D*, *Figure 5—figure supplement 1A-H*). Vehicle-treated control animals showed only typical short-distance sprouting of serotonergic axons less than 2 mm distal to the lesion site (*Figure 5E–G and K*). At the same time, DMAPT treatment increased the number and length of axons, with the longest reaching 4 mm (*Figure 5E and H–K*). When analyzed 8 weeks after SCC (*Figure 6A*), DMAPT further accelerated raphespinal regeneration, with the longest axons reaching >7 mm (*Figure 6B–M*). However, despite the more pronounced anatomical regeneration, the BMS score was only slightly further increased compared to 14 days after SCC (*Figure 6N*). In addition, the lesion size and CSPG expression at the glial scar remained unaffected by DMAPT treatment (*Figure 6—figure supplement 1A-D*). Thus, systemic DMAPT application accelerates raphespinal regeneration and some functional recovery after a severe spinal cord injury.

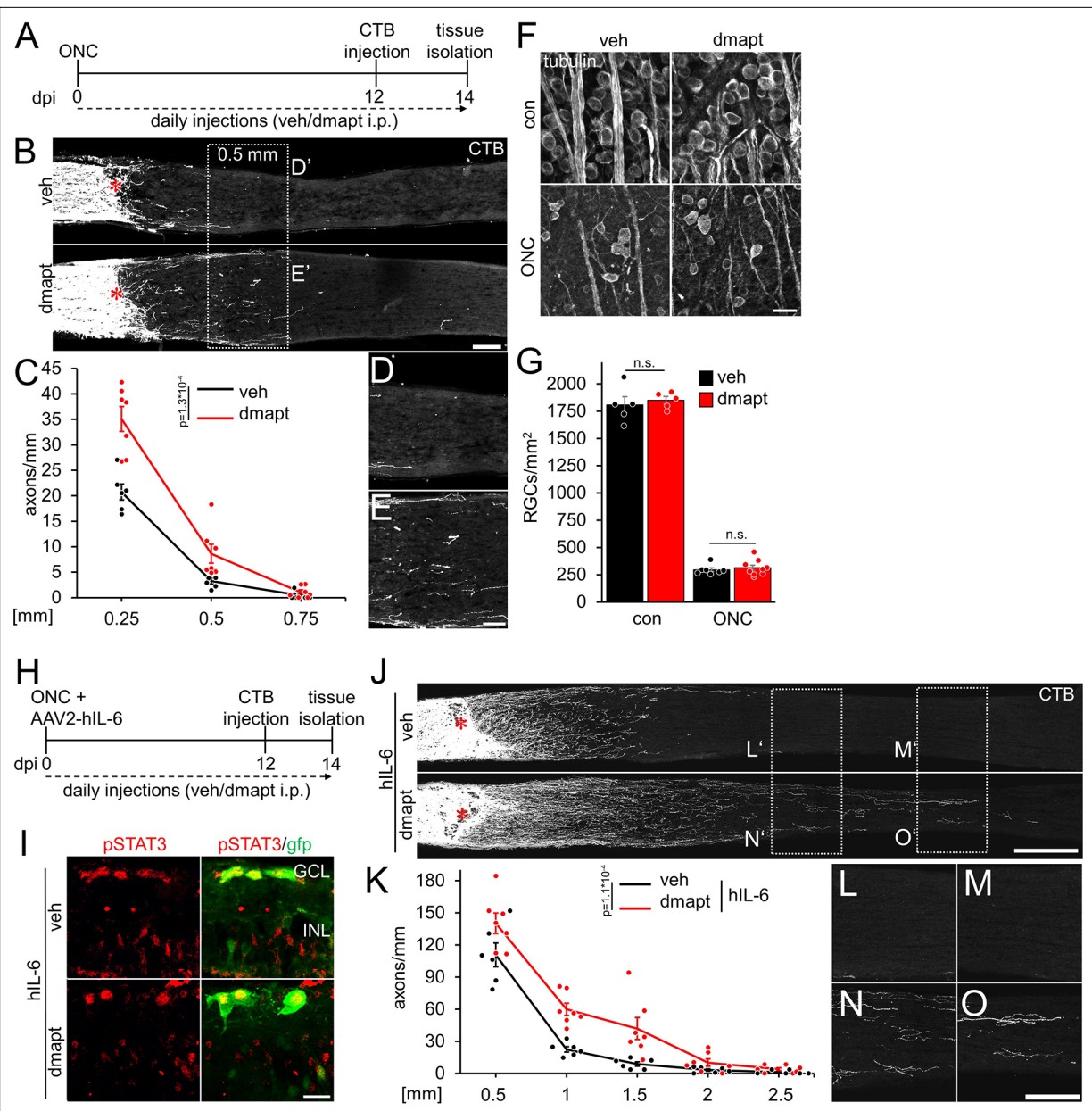

**Figure 4.** DMAPT promotes optic nerve regeneration and enhances the effect of hIL-6. (**A**) Timeline showing surgical interventions and intraperitoneal (i.p.) drug application for experiments presented in **B–G**. (**B**) Longitudinal optic nerve sections containing CTB (white) traced axons from mice 2 weeks after optic nerve crush (ONC) and daily repeated intraperitoneal vehicle (veh) or DMAPT injections, as described in **A**. Red asterisks indicate the lesion sites. Scale bar: 100 µm. (**C**) Quantification of regenerating axons at indicated distances distal to the lesion site in optic nerves as depicted in **B**, showing significantly enhanced optic nerve regeneration after DMAPT treatment compared to controls. Values represent the mean ± SEM of 6–7 animals per group (veh: n=6, dmapt: n=7). (**D, E**) Higher magnification of dashed boxes as indicated in **B**. Scale bar: 50 µm. (**F**) Confocal scans of retinal wholemounts from mice either uninjured (con) or subjected to ONC 14 days before tissue isolation. Animals received daily intraperitoneal injections of vehicle (veh) or DMAPT. RGCs were visualized by immunohistochemical βIII-tubulin staining (tubulin). Scale bar: 20 µm. (**G**) Quantification of RGC density in retinal wholemounts, as described in **F**. DMAPT did not affect RGC survival in uninjured mice or after ONC. Values represent the mean ± SEM of 5–9 animals per group (veh, con: n=5; dmapt, con: n=5; veh ONC: n=7; dmapt, ONC: n=9). (**H**) Timeline showing surgical interventions and intraperitoneal (i.p.) drug application for experiments presented in **I–K** and *Figure 3—figure supplement 1*. (**I**) Immunostained retinal cross-sections prepared from mice that had received an intravitreal injection of AAV2-hIL-6-GFP simultaneously to optic nerve crush (ONC) and then daily intraperitoneal injections of vehicle or DMAPT (2 µg/kg) for 2 weeks. Sections were immunostained for phosphorylated STAT3 (pSTAT3, red). Transduction with AAV-2 hIL-6 is visualized by GFP (green) staining. hIL-6-induced STAT3-phosphorylation was not affected by DMAPT cotreatment. Scale bar: 20 µm. (**J**) Longitudinal optic nerve sections containing CTB (white) traced axons from mice described in **H**. Red asterisks indicate the lesion sites. Scale bar: 200 µm. (**K**) Quantification of regenerating axons distal to the lesion site at indicated distances in optic nerves from mice that received AAV2-hIL-6 combined with

*Figure 4 continued on next page*

*Figure 4 continued*

vehicle or DMAPT treatment, as shown in **J**. DMAPT significantly enhanced hIL-6-induced optic nerve regeneration. Values represent the mean ± SEM of 6–7 animals per group (veh: n=6; dmapt: n=7). (**L–O**) Higher magnification of dashed boxes as indicated in **J**. Scale bar: 100 µm. Significances of intergroup differences in **C**, **G**, and **K** were evaluated using a two-way analysis of variance (ANOVA) with a Tukey post hoc test. p-Values indicate statistical significance. Dots represent values from single animals.

The online version of this article includes the following figure supplement(s) for figure 4:

**Figure supplement 1.** Optic nerve regeneration after AAV2-hIL-6 application and DMAPT treatment at suboptimal concentrations.

## Raphespinal regeneration contributes to DMAPT-mediated functional recovery

We prepared additional mice to investigate whether the DMAPT-mediated effects on functional recovery depend on serotonergic fiber regeneration, as previously shown for hIL-6 (*Leibinger et al., 2021*). To this end, we performed SCC with either vehicle or DMAPT treatment. After the DMAPT group had reached maximum functional recovery, we injected the serotonergic neurotoxin 5,7-dihydroxytryptamine (DHT), which depletes raphe neurons as early as 1 day after injection, as shown previously (*Kim et al., 2004*; *Leibinger et al., 2021*). Expectedly, DHT treatment reduced the

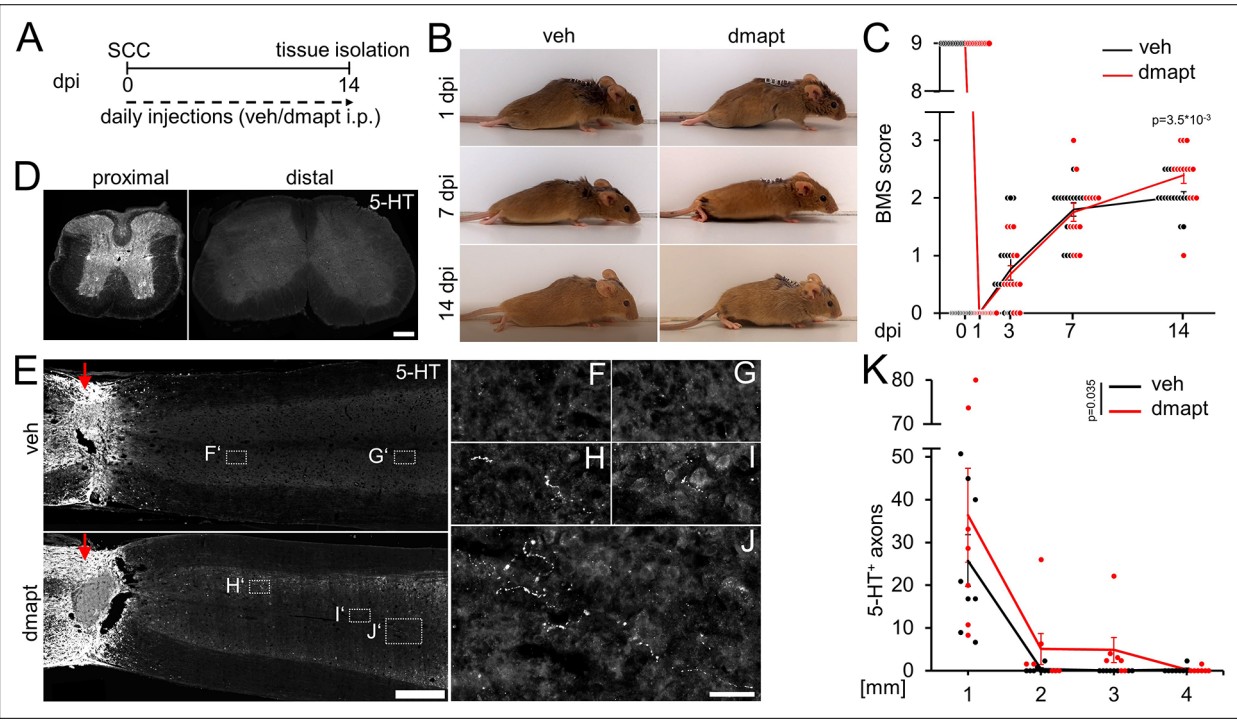

**Figure 5.** DMAPT accelerates RpST regeneration and functional recovery after spinal cord crush. (**A**) Timeline showing surgical interventions, BMS testing, and intraperitoneal (i.p.) drug application for experiments presented in **B–K**. (**B**) Representative photos showing open field hindlimb movement of mice at 1, 7, and 14 days post-injury (dpi). Before, mice had received spinal cord crush (SCC) and daily intraperitoneal injections of either DMAPT (dmapt) or the vehicle (veh). (**C**) BMS score of animals as described in **A** and **B** over 2 weeks after SCC. DMAPT significantly improved hindlimb movement's functional recovery, resulting in plantar paw placement in most animals 2 weeks after spinal cord crush. Values represent the mean ± SEM of 13–17 animals per group (veh: n=13; dmapt n=17). (**D**) Immunohistochemical staining of transverse spinal cord sections from mice treated as described in **A** and **B** 2 weeks after injury. Serotonergic (5-HT) axons are present in the raphespinal tract proximal but not distal to the lesion site, indicating lesion completeness. Scale bar: 200 µm (**E**) Coronal thoracic spinal cord sections isolated from mice 2 weeks after SCC and daily intraperitoneal injections of either dmapt or the veh. Raphespinal tract (RpST) axons were stained using an anti-serotonin antibody (5-HT, white). Only dmapt but not vehicle-treated mice showed regeneration of serotonergic axons >2 mm beyond the lesion site (red arrow). Scale bar: 500 µm. (**F–J**) Higher magnifications of dashed boxes as indicated in **E**. Scale bar: 50 µm. (**K**) Quantification of regenerating 5-HT-positive axons from animals described in **A** and **E** at indicated distances beyond the lesion. Values represent the mean ± SEM of 7–8 mice per group (veh: n=8; dmapt: n=7). Significances of intergroup differences in **C** and **K** were evaluated using a two-way analysis of variance (ANOVA) with a Tukey post-hoc-test. p-Values indicate statistical significance.

The online version of this article includes the following figure supplement(s) for figure 5:

**Figure supplement 1.** Verification of spinal cord lesion completeness.

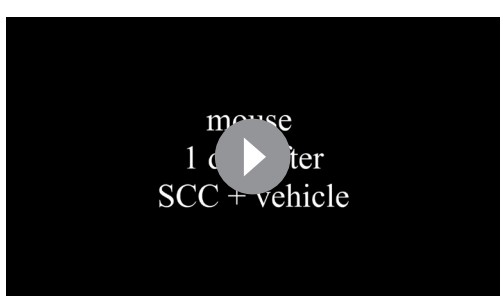

**Video 1.** Open field locomotion of vehicle-treated control mouse. The video shows the same mouse 1 day, 1 week and 2 weeks after spinal cord crush (SCC). One day post SCC there is no hindlimb movement. After 2 weeks, the mouse has recovered from the spinal shock and performs ankle movement (BMS score 2). https://elifesciences.org/articles/88279/figures#video1

number of serotonergic brainstem raphe neurons and their axons without affecting other neuronal populations (*Figure 7A–J*; *Figure 7—figure supplement 1A,B*). This did not affect BMS scores of vehicle-treated controls (*Figure 7K and L*; *Video 3*) but immediately abolished DMAPT-induced gain of motor function (*Figure 7K and L*; *Video 4*), suggesting an essential contribution of regenerated serotonergic axons.

## DMAPT accelerates hIL-6-induced functional recovery after complete spinal cord crush

Intracortical injection of AAV2-hIL-6 promotes axon regeneration and functional recovery after SCC (*Leibinger et al., 2021*). To test for synergistic effects, as seen in the optic nerve (*Figure 4*), we combined hIL-6 and DMAPT treatment. To this end, we subjected wild-type mice to SCC and injected AAV2-hIL-6 into the sensorimotor cortex. Mice received daily intraperitoneal injections of either vehicle or DMAPT for 8 weeks (*Figure 8A*). As determined by immunostaining, AAV2-hIL-6-mediated transduction of layer V cortical motorneurons induced robust phosphorylation of STAT3 (*Figure 8—figure supplement 1A-E*). Consistent with (*Leibinger et al., 2021*), intracortical AAV2-hIL-6 also transneuronally stimulated medullary raphe neurons (*Figure 8—figure supplement 1I-M*). At the same time, DMAPT treatment did not affect STAT3 phosphorylation in cortical (*Figure 8—figure supplement 1B,F,G,H*) or raphespinal (*Figure 8—figure supplement 1J,N,O,P*) neurons at applied doses used in this study.

Consistent with previous results (*Leibinger et al., 2021*), AAV2-hIL-6 treatment restored plantar hind paw placement and stepping with lift-off, followed by forwarding limb advancement and re-establishment of bodyweight support, reaching a maximum BMS score of 4–5 (*Figure 8B–C*). Interestingly, combinatorial treatment with DMAPT accelerated functional recovery, allowing plantar steps in most animals already 14 d after SCC. These mice reached the maximum effect at 21 days after injury, while animals treated with hIL-6 needed 42 days (*Figure 8B–C*; *Videos 5 and 6*). Moreover, additional DMAPT treatment also reduced the variance, with fewer weakly responding animals, resulting in a higher average final score of 4 compared to 3.5 in the hIL-6 control group.

## DMAPT affects RpST fiber regeneration after spinal cord crush

Regarding fiber regeneration, additional DMAPT treatment further enhanced the hIL6-induced regeneration of serotonergic fibers 8 weeks after SCC (*Figure 8D–K*). Compared to AAV2-hIL-6 therapy, which led to robust axonal regeneration up to 7 mm distal from the lesion site (comparable to *Leibinger et al., 2021*; *Figure 8D-G,K*), additional DMAPT treatment significantly increased the number of serotonergic axons at this distance, with some axons reaching even 8 mm (*Figure 8D and H–K*).

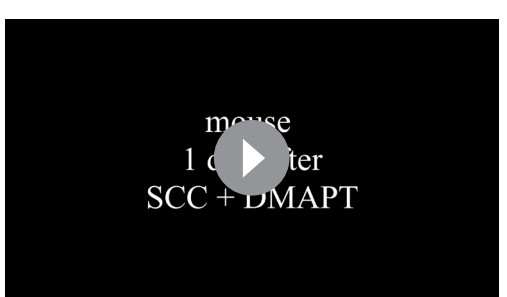

**Video 2.** Open field locomotion of DMAPT-treated mouse. The video shows the same mouse 1 day, 1 week and 2 weeks after spinal cord crush (SCC). One day post SCC there is no hindlimb movement, whereas after 2 weeks the mouse shows functional recovery of plantar paw placement with support of the bodyweight (BMS score 3). https://elifesciences.org/articles/88279/figures#video2

## Discussion

Here, we demonstrate that the systemic application of DMAPT facilitates axon regeneration in the injured optic nerve and accelerates functional recovery after severe spinal cord injury. Parthenolide's effects on CNS regeneration *in vivo* were confirmed in cell culture using primary RGCs from

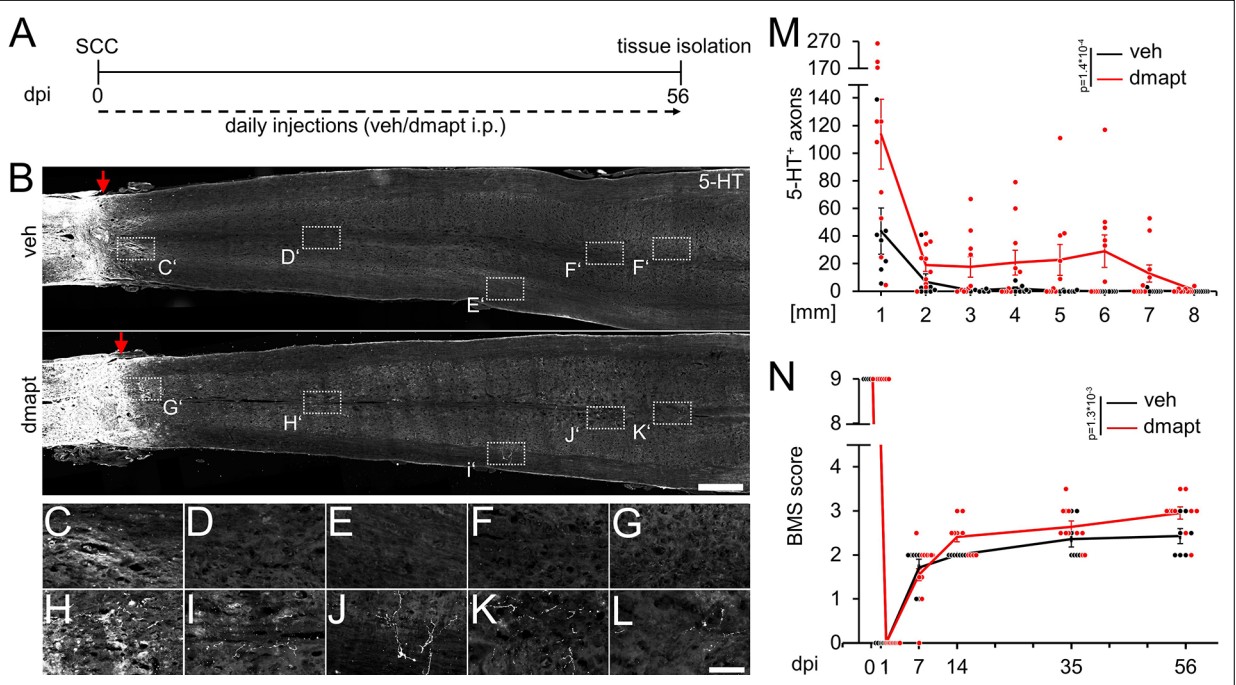

**Figure 6.** Spinal cord regeneration and functional recovery after long-term DMAPT treatment. (**A**) Timeline showing surgical interventions and intraperitoneal (i.p.) drug application for experiments presented in **B–N**. (**B**) Coronal thoracic spinal cord sections isolated from mice 8 weeks after spinal cord crush (SCC) and daily intraperitoneal injections of either DMAPT (dmapt) or the vehicle (veh). Raphespinal tract (RpST) axons were stained using an anti-serotonin antibody (5-HT, white). In contrast to the vehicle-treated control, dmapt enabled RpST regeneration over long distances beyond the lesion site (red arrow). Scale bar: 500 µm. (**C–L**) Higher magnification of dashed boxes indicated in **B**. Scale bar: 50 µm. (**M**) Quantification of regenerating 5-HT-positive axons as described in **A** and **B** at indicated distances beyond the lesion. Values represent the mean ± SEM of 7–10 animals per group (veh: n=7; DMAPT: n=10). (**N**) BMS score of animals as described in **A** and **B**, over 8 weeks after spinal cord injury. DMAPT significantly improved functional recovery of hindlimb movement, reaching a plateau 2 weeks after SCC. Values represent the mean ± SEM of 7–10 animals per group (veh: n=7; DMAPT: n=10). Significances of intergroup differences in **M** and **N** were evaluated using a two-way analysis of variance (ANOVA) with a Tukey post-hoc-test. p-Values indicate statistical significance.

The online version of this article includes the following figure supplement(s) for figure 6:

**Figure supplement 1.** DMAPT does not affect lesion size or CSPG release after SCC.

adult mice and humans. As on sensory neurons (preprint server, *Gobrecht et al., 2022*), parthenolide reduces microtubule detyrosination in axonal tips, which increases their dynamics and promotes axon extension (*Gobrecht et al., 2016*). As for sensory neurons, this effect is likely mediated by inhibiting VASH1/VASH2, the enzymes catalyzing the detyrosination (*Gobrecht et al., 2022*). Consistently, the effective concentration range and doses of parthenolide/DMAPT to accelerate axon regeneration of RGCs and serotonergic fibers in the spinal cord acompanied with functional recovery was similar to sensory neurons and regeneration in the injured peripheral nerve (*Gobrecht et al., 2022*). Parthenolide did not affect CRMP2 or STAT3 activity at the nanomolar concentrations used in this study, both critical for CNS axon regeneration (*Leibinger et al., 2016*; *Leibinger et al., 2019*), suggesting the reduction of microtubule detyrosination causes the axon growth-promoting effects. Although we cannot exclude the possibility that other known activities of parthenolide/DMAPT, such as oxidative stress or NF-kB inhibition, could have contributed to the observed effects, this is rather unlikely because such effects have only been reported at much higher micromolar concentrations (*Bork et al., 1997*; *Saadane et al., 2007*; *Carlisi et al., 2016*; *Gobrecht et al., 2016*).

Systemic DMAPT treatment significantly promotes axon extension in the optic nerve and regeneration of serotonergic fibers beyond the injury site. The selective elimination of serotonergic fibers by DHT (*Kim et al., 2004*; *Capela et al., 2008*; *Cummings et al., 2009*) abolishes the recovered locomotion, demonstrating that the improved regeneration of serotonergic fibers is essential for recovery after SCC. These data are consistent with previously published data, showing a similar effect on AAV-hIL-6-mediated recovery (*Leibinger et al., 2021*). It might appear conceivable that the depletion of

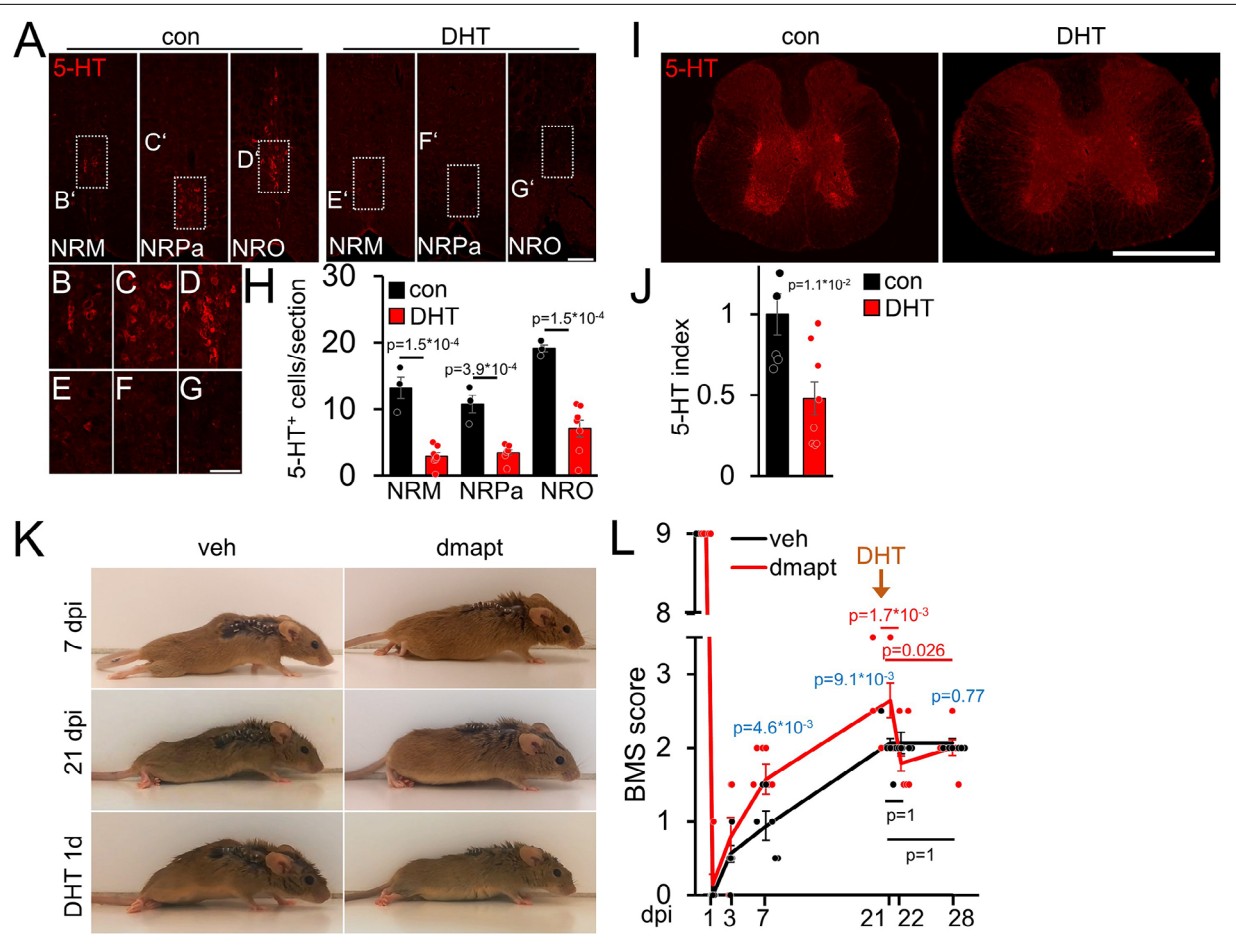

**Figure 7.** DMAPT-mediated functional recovery is RpST-dependent. (**A**) Validation of DHT-mediated depletion of serotonergic (5-HT, red) neurons in raphe nuclei (NRM = nucleus raphe magnus, NRPa = nucleus raphe pallidus, NRO = nucleus raphe obscurus) located above the pyramidal tracts in coronal medullary sections from mice as depicted in **K, L** 1 week after intracerebroventricular 5, 7-dihydroxytryptamine (DHT) injection, or untreated controls (con). Scale bar: 100 µm. (**B–G**) Higher magnification of indicated areas in images, as shown in **A**. Scale bar: 50 µm. (**H**) Quantification of serotonin (5-HT) positive raphe neurons per section in medullary immunostainings for untreated (con: n=5) and DHT-injected (DHT: n=8) mice as described in **A**. (**I**) Thoracic spinal cord cross-sections from animals, as described in **A**. Raphespinal axons were visualized by serotonin (5-HT) staining. Scale bar: 500 µm. (**J**) Quantification of 5-HT staining as shown in **I** for untreated (con: n=5) and DHT-injected (DHT, n=8) mice. (**K**) Representative images of vehicle- (veh) or DMAPT- (dmapt) treated mice at 7 or 21 d post-injury (dpi) and 1 day after DHT treatment. (**L**) BMS score of animals treated as described in (**K**) at indicated time points after spinal cord crush and DHT treatment. Values represent means ± SEM of 7–8 animals per group (veh: n = 8; dmapt: n=7), showing the average left and right hind paws score. Significances of intergroup differences were evaluated using a two-way analysis of variance (ANOVA) with a Holm Sidak post hoc test (**H, L**) or Student's t-test (**J**). The significance between group means is indicated by p-values. In **L**, comparisons are shown within the vehicle group (black), within the dmapt group (red), and between groups (blue).

The online version of this article includes the following figure supplement(s) for figure 7:

**Figure supplement 1.** DHT specifically depletes serotonergic neurons.

non-regenerated serotonergic axons may have contributed to these results. However, we can largely rule this out since DHT did not influence the injured but non-regenerated vehicle control group. Furthermore, we have shown in a previous publication that the general depletion of serotonergic neurons in uninjured animals also has no significant influence on open-field locomotion as measured in the BMS score and subscore (*Leibinger et al., 2021*).

Moreover, even though DMAPT-mediated functional improvement depends on the regeneration of serotonergic fibers, we cannot exclude the possibility that spinal tracts other than the raphespinal tract also contributed. The fact that DMAPT improved axon regeneration in the optic nerve and the spinal cord also implies that sufficient amounts of the drug reached the axonal growth cones, where the drug affects the microtubule detyrosination (*Gobrecht et al., 2016*). It is likely that an oral

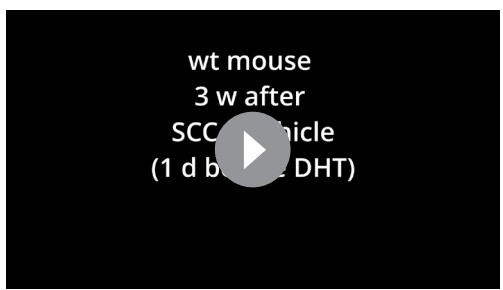

**Video 3.** Open field locomotion of vehicle-treated control mouse 3 w after SCC before and after depletion of serotonergic neurons with DHT. The mouse shows ankle movement (BMS score 2) 3 weeks after SCC, which is not impaired by DHT treatment.
https://elifesciences.org/articles/88279/figures#video3

application of DMAPT, which reportedly is >60% bioavailable (*Guzman et al., 2007*; *Song et al., 2014*; *Gobrecht et al., 2022*), has similar effects on nerve regeneration, bolstering the potential as a drug to treat CNS axonal damage. In contrast to the raphespinal tract, axon regeneration in the optic nerve after sole DMAPT treatment might appear only moderate. However, in contrast to raphe neurons, RGCs are subject to apoptotic cell death after axotomy, and DMAPT acts on the axonal growth cones without mediating neuro-protection. Thus, DMAPT can only develop its full effects in combination with neuroprotective approaches like hIL-6, which activates cell-intrinsic signaling pathways and the survival of a larger number of RGCs. In this combination, DMAPT dramatically enhanced axon regeneration.

Another significant finding of the current study is that pro-regenerative hIL-6 and *Pten*[-/-] treatments induce microtubule detyrosination in RGC axons. Given that a parthenolide-mediated reduction of microtubule detyrosination promotes axon growth, these observations suggest that this effect of hIL-6 or *Pten*[-/-] compromises their full axon regeneration-promoting potential. hIL-6 and *Pten*[-/-] induced microtubule detyrosination is likely caused by inhibitory GSK3 phosphorylation via increased PI3K/AKT activity. As a result, MAP1B activity is decreased, and the axonal detyrosination rate is elevated (*Guo et al., 2016*; *Leibinger et al., 2017*; *Kath et al., 2018*). However, CNTF does not affect detyros-ination due to its lower AKT activation than hIL-6 (*Müller et al., 2007*; *Müller et al., 2009*; *Leibinger et al., 2016*). At the same time, hIL-6 and *Pten*[-/-] mediated GSK3 inhibition confers disinhibitory effects toward myelin-associated molecules because CRMP2 is less inhibited (*Liz et al., 2014*; *Leibinger et al., 2017*; *Leibinger et al., 2019*). Thus, the disinhibitory effect of GSK3 inhibition promotes regeneration, while the detyrosinating effect on microtubules rather compromises it. Therefore, by counteracting detyrosination, parthenolide/DMAPT cotreatment further increases the axon growth-promoting effect of hIL-6. Consistently, effective parthenolide concentrations reduce detyrosination levels in hIL-6-treated, and *Pten*[-/-] RGCs, and combined treatment leads to more substantial neurite growth than each treatment alone. This notion is also supported by the fact that *Pten*[-/-] and hIL-6-stimulated neurons sustain parthenolide-induced additive elevation of axon growth even at higher parthenolide concentrations, which already diminish outgrowth on naïve or CNTF-stimulated RGCs. The level of detyrosinated tubulin at the maximal applied parthenolide concentration in hIL-6 treated RGCs was still similar to untreated control RGCs. The synergistic effects of DMAPT and hIL-6 were also reproduced *in vivo* in the optic nerve and spinal cord. Thus, parthenolide/DMAPT neutralizes the hIL-6-mediated limiting effect of decreased microtubule dynamics and synergistically acceler-ates CNS regeneration.

Our findings also indicate that DMAPT mainly accelerates axon growth and has little effect on the intrinsic regenerative state of neurons. There-fore, the RGC regeneration-promoting effects of parthenolide on its own are only moderate but synergistically increase hIL-6-mediated axon extension. Nevertheless, DMAPT enabled substantial regeneration of serotonergic RpST after complete SCC. Furthermore, the length of regenerating RpST axons after SCC further increased from 2 to 8 wpc, while the functional recovery was already almost maximal in the early phase. This observation might be based on the different possibilities of achieving functional

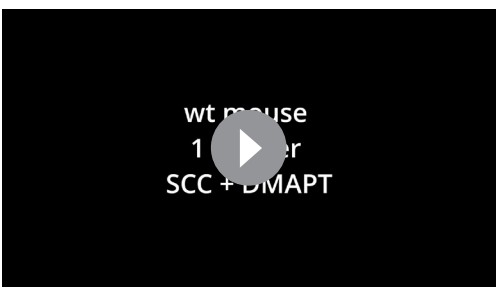

**Video 4.** Open field locomotion of DMAPT-treated mouse 1 w and 3 w after SCC, and after depletion of serotonergic neurons with DHT. The mouse shows ankle movement 1 w after SCC and functional recovery of plantar paw placement with support of the bodyweight after 3 w (BMS score 3), which is abolished after DHT treatment (BMS score 2).
https://elifesciences.org/articles/88279/figures#video4

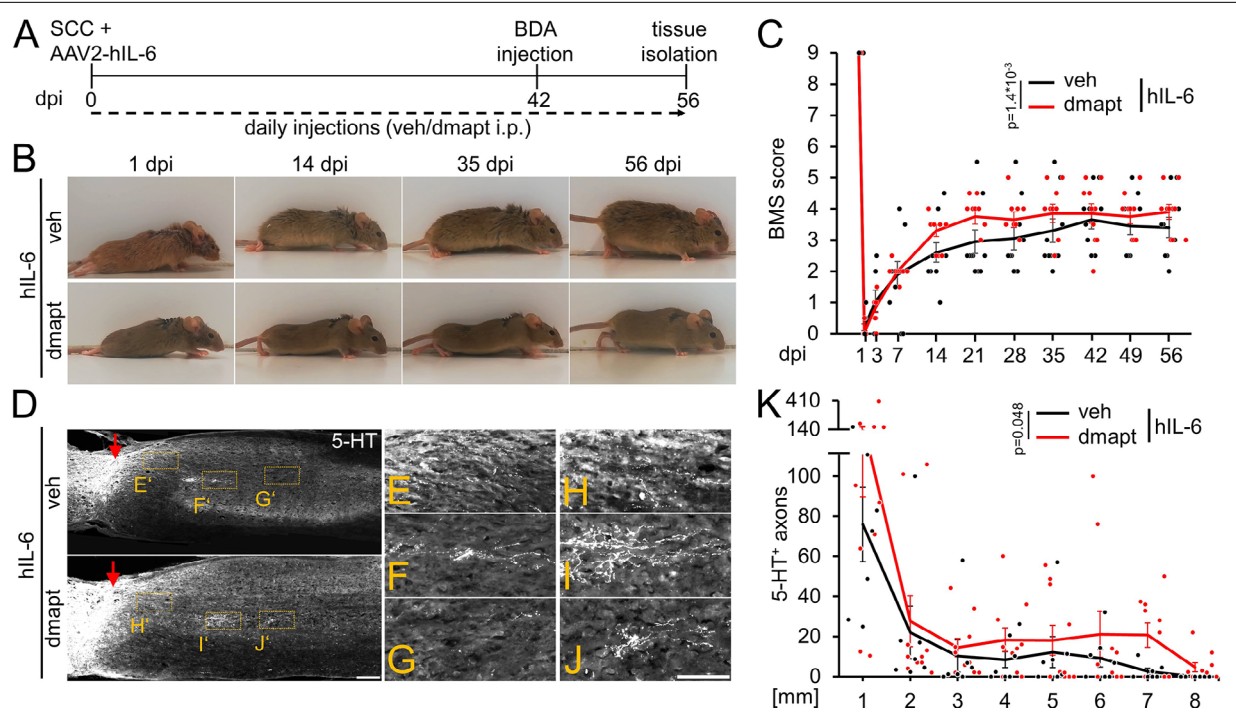

**Figure 8.** DMAPT treatment accelerates hIL-6-induced RpST regeneration and functional recovery after spinal cord crush. (**A**) Timeline showing surgical interventions and intraperitoneal (i.p.) drug application for experiments presented in **B–K**. (**B**) Representative photos showing open field hindlimb movement of mice at 1, 14, 35, and 56 days post-injury (dpi) of animals as described in **A**. (**C**) BMS scores of animals as described in **A** over 8 weeks after SCC. DMAPT treatment accelerated functional recovery of hindlimb movement in hIL-6-treated mice. Values represent the mean ± SEM of 10 animals per group (veh: n=10; DMAPT: n=10). (**D**) Sagittal thoracic spinal cord sections isolated from mice described in **A**, 8 weeks after spinal cord crush (SCC), followed by intracortical AAV2-hIL-6 injection and daily intraperitoneal injection of either vehicle (veh) or DMAPT (dmapt). Raphespinal tract axons were stained using an anti-serotonin antibody (5-HT, white). DMAPT treatment did not significantly affect axon regeneration compared to vehicle-treated controls. Scale bar: 250 μm (**E–J**) Higher magnifications of dashed boxes as indicated in **D**. Scale bar: 100 μm (**K**) Quantification of regenerated 5-HT positive axons from animals described in **A** at indicated distances beyond the lesion site. DMAPT did not significantly enhance hIL-6-mediated raphespinal axon regeneration further. Values represent the mean ± SEM of 7–10 animals per group (veh: n=7; DMAPT: n=10). Values represent means ± SEM of 7 or 10 (veh: n=7; dmapt: n=10) animals per group. Dots represent values for single animals. Significances of intergroup differences in **C**, and **K** were evaluated using a two-way analysis of variance (ANOVA) with a Tukey post-hoc- test. p-Values indicate statistical significance.

The online version of this article includes the following figure supplement(s) for figure 8:

**Figure supplement 1.** DMAPT does not affect hIL-6-mediated STAT3 activation.

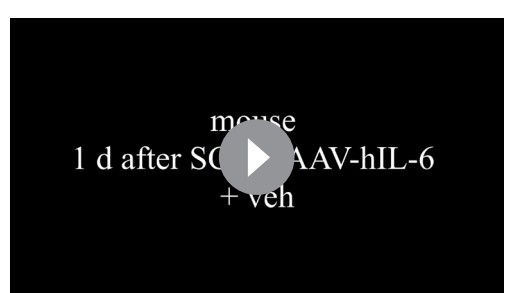

**Video 5.** Open field locomotion of vehicle-treated mouse 1 day, two weeks and 5 weeks after AAV-hIL-6 treatmen. HIL-6 treatment induces functional recovery. https://elifesciences.org/articles/88279/figures#video5

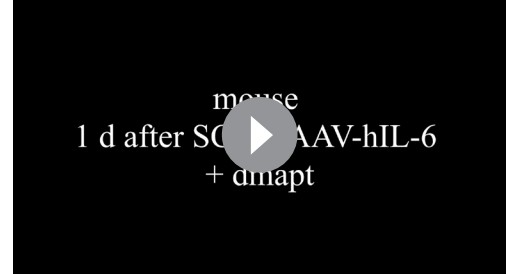

**Video 6.** Open field locomotion of DMAPT-treated mouse 1 day, 2 weeks and 5 weeks after AAV-hIL-6 treatment. DMAPT accelerates hIL-6-induced functional recovery. https://elifesciences.org/articles/88279/figures#video6

recovery. On the one hand, descending supraspinal axons might need to grow over long distances to reestablish functional connections within the lumbar central pattern generator (CPG), a network of neurons responsible for generating stereotyped hindlimb movement (*Courtine et al., 2008*; *van den Brand et al., 2012*; *Kiehn, 2016*). On the other hand, it is also possible that severed axons only need to overcome short distances of a few spinal segments and then connect to existing propriospinal interneurons, which might serve as a relay station to the CPG (*Courtine et al., 2008*; *Anderson et al., 2018*; *Chen et al., 2018*). An observation accounting for the latter possibility is increased axonal sprouting in the region of the lumbar CPG (~6–7 mm distal to the lesion) at 8 wpc, whereas the substantial increase in functional recovery occurred already earlier (~2–3 wpc) before the axons had reached the CPG. Thus, DMAPT-induced long-distance RpST regeneration might not be essential for its beneficial effects on functional recovery, whereas DHT experiments showed that RpST regeneration, in general, is crucial.

In conclusion, systemically applied DMAPT measurably accelerates CNS regeneration and functional recovery and overcomes the limiting effect of AKT/GSK3 on elevating microtubule detyrosination. Whether DMAPT treatment can also improve functional recovery in more clinically relevant injury models, such as contusion injuries, or even more effective in combination with other regeneration-promoting strategies (*Fischer et al., 2004*; *Lang et al., 2015*; *Burnside et al., 2018*) will be investigated in the future. Finally, the axon growth-promoting effects on primary adult human CNS neurons underline the potential of parthenolide/DMAPT as a possible future treatment strategy for CNS injury in humans.

## Materials and methods

### Mouse strains

Male and female C57BL/6 and 129/Ola mice were used for all experiments. All mice were housed under the same conditions for at least 10 days before the experiments and maintained on a 12 hr light/dark cycle with *ad libitum* access to food and water. All experimental procedures were approved by the local animal care committee (LANUV Recklinghausen; reference number of approval: 84–02.04.2017 . A218, 84–02.04.2021.A142) and were conducted in compliance with federal and state guidelines for animal experiments in Germany.

### Human eyes

The use of human eyes and publishing of the obtained results was approved by the ethics committee of Heinrich Heine University Düsseldorf (study number 4067) and performed by the ethical standards as laid down in the 1964 Declaration of Helsinki and its later amendments or comparable ethical standards. Informed consent, and consent to publish, was obtained.

### Surgical procedures

#### Optic nerve crush and intravitreal injections

Adult C57BL/6 mice were anesthetized by intraperitoneal injections of ketamine (120 mg/kg) and xylazine (16 mg/kg). For western blot and cell culture experiments, RGCs were transduced 3 weeks before tissue isolation by intravitreal injection of 2 µl AAV2-hIL-6. For *in vivo* regeneration experiments, the left optic nerve was intraorbitally crushed 1 mm behind the eyeball for 10 s using a jeweler's forceps (Hermle), as described previously (*Leibinger et al., 2009*). Mice received intravitreal injections of 2 µl AAV2-hIL-6 directly after optic nerve crush. Two days before tissue isolation, regenerating axons were anterogradely labeled by intravitreal injection of 2 µl Alexa Fluor 555-conjugated cholera toxin β subunit (CTB, 0.5% in PBS; Invitrogen).

#### Intracortical viral injections

Adult 129/Ola mice were placed in a stereotactic frame and continuously supplied with 1.5–2% isoflurane for anesthesia. A midline incision was made over the skull to open the skin and reveal bregma. A micro drill with a 0.5 mm bit was used to open a 2x2 mm window on each side of the skull to expose the sensorimotor cortex. AAV2-hIL-6 was injected into the cortex layer V to transduce corticospinal neurons. Four injections of 500 nl each were applied into the left cortex through a glass capillary attached to a nanoliter injector (Nanoject III, Drummond) at the following coordinates: 1.5 mm lateral,

0.6 mm deep, and 0.5 mm anterior; 0.0 mm, 0.5 mm, and 1.0 mm posterior to bregma. At each injection site, five pulses of 100 nl were applied at a rate of 10 nl/s. A pause of 10 s was made after each pulse to allow the distribution of the virus solution. The needle was left in place for 1 min before moving to the next site.

## Complete spinal cord crush

A complete spinal cord crush was performed as described previously (*Leibinger et al., 2021*; *Terheyden-Keighley et al., 2023*). Adult 129/Ola mice were continuously supplied with 1.5–2% isoflurane for anesthesia. A midline incision of ~1.5 cm was made over the thoracic vertebrae. The fat and muscle tissue were cleared from thoracic vertebrae 7 and 8 (T7, T8). While holding onto T7 with forceps, a laminectomy was performed at T8 to expose the spinal cord. The complete spinal cord was then crushed for 2 s with forceps that had been filed to a width of 0.15 mm for the last 5 mm of the tips to generate a homogeneously thin lesion site (*Zukor et al., 2013*). After surgery, muscle layers were sutured with 5.0 resorbable sutures, and the skin was secured with wound clips.

## DHT injection

To deplete serotonergic inputs to the spinal cord, mice received bilateral intracerebroventricular injections of 30 µg of the serotonin neurotoxin 5, 7-dihydroxytryptamine (DHT, Biomol). DHT was dissolved in 0.5 µl of 0.2% ascorbic acid in saline, as described previously (*Leibinger et al., 2021*). To this end, mice were anesthetized and fixed under a stereotactic frame described above for intracortical injections. In addition, the tip of a glass micropipette attached to a nanoject III (Drummond) was positioned at the following coordinates: 0.6 mm posterior, 1.6 mm lateral to bregma, and 2 mm deep from the cortical surface. DHT was injected at the same rate as AAV2 injections, and the pipette was left for 1 min before the withdrawal. Thirty minutes before the DHT injection, the monoamine uptake inhibitor desipramine (Sigma) was intraperitoneally administered at 25 mg/kg to prevent DHT uptake into noradrenergic neurons.

## Preparation of AAV2

AAV plasmids carrying cDNA for hIL-6 downstream of a CMV promoter were co-transfected with pAAV-RC (Stratagene) encoding the AAV genes rep and cap, and the helper plasmid (Stratagene) encoding E24, E4, and VA into AAV-293 cells (Stratagene) for recombinant AAV2 generation. In addition, purifying virus particles was performed as described previously (*Zolotukhin et al., 1999*; *Leibinger et al., 2016*; *Leibinger et al., 2021*).

## Western blot assays

For protein lysate preparation, optic nerves from adult C57BL/6 mice, either untreated or 3 weeks after intraperitoneal AAV2-hIL-6 injection, were dissected and homogenized in lysis buffer (20 mM Tris/HCl pH 7.5, 10 mM KCl, 250 mM sucrose, 10 mM NaF, 1 mM DTT, 0.1 mM Na3VO4, 1% Triton X-100, 0.1% SDS) with protease inhibitors (Calbiochem, Darmstadt, Germany) and phosphatase inhibitors (Roche, Basel, Switzerland) using 5 sonication pulses at 40% power (Bandelin Sonoplus, Berlin, Germany). Lysates were cleared by centrifugation in an Eppendorf tabletop centrifuge at 4150×g for 10 min at 4 °C. Proteins were separated by sodium-dodecyl-sulfate-polyacrylamide gel electrophoresis (SDS-PAGE) using Mini TGX gels (10%, Bio-Rad, Hercules, USA) according to standard protocols and transferred to nitrocellulose membranes (0.2 µm, Bio-Rad, Hercules, USA). Blots were blocked in 5% dried milk in phosphate-buffered saline with 0.05% Tween-20 (PBS-T; Sigma, St Louis, USA) and incubated with antibodies against S21-phosphorylated GSK3α (1:2000; Cell Signaling Technology, RRID: AB_659836), S9-phosphorylated GSK3β (1:1000; Cell Signaling Technology, RRID:AB_2115196), total GSK3α/β (1:1000; Cell Signaling Technology, RRID: AB_10547140), T514-phosphorylated CRMP2 (1:1000; Abcam, RRID:AB_942229), total CRMP2 (1:1000; Cell Signaling Technology, RRID:AB_2094339), detyrosinated tubulin (1:1000; Millipore, RRID:AB_177350) and βIII-tubulin (1:2000; BioLegend, RRID:AB_2313773) at 4 °C overnight. All antibodies were diluted in PBS-T containing 5% BSA (Sigma). A conformation-specific HRP-conjugated anti-rabbit IgG (1:4000; Cell Signaling Technology) was used as a secondary antibody. Antigen-antibody complexes were visualized using enhanced chemiluminescence substrate (Bio-Rad) on a FluorChem E detection system

(ProteinSimple). Western blots were repeated at least three times to verify the results. Band intensities were quantified relative to respective loading controls by using ImageJ software.

## Dissociated mouse retinal cell cultures

Adult wildtype C57BL/6 mice or Pten-floxed mice (C57BL/6;129/J-TgH(Pten-flox)) were used for all cell culture experiments. Some experiments included animals that had received intravitreal AAV2-hIL-6 injections 3 weeks before dissociation. Retinal cultures were prepared as described previously (*Grozdanov et al., 2010*). In brief, retinae were dissected from the eyecups and digested in Dulbecco's Modified Eagle medium (DMEM; Thermo Fisher) containing papain (10 U/ml; Worthington) and L-cysteine (0.2 μg/ml; Sigma) at 37 °C for 30 min. After digestion, retinae were washed with DMEM, triturated, and centrifuged in 50 ml DMEM (500 g, 7 min). Cell pellets were resuspended in a medium containing B27 supplement (1:50; Invitrogen) and penicillin/streptomycin (1:50; Merck) and passed through a cell strainer (40 μm; Greiner Bio-One). Cells were treated with parthenolide (Sigma; 0.25–5.0 nM in dimethyl sulfoxide), recombinant rat CNTF (Peprotech; 200 ng/ml), DMAPT (Abcam; 0.25–5.0 nM in dimethyl sulfoxide), or a combination of parthenolide or CNTF. 300 μl cell suspension was added to each well on 4-well plates (Nunc), which were coated with poly-D-lysine (0.1 mg/ml, molecular weight 70,000–150,000 Da; Sigma) and laminin (20 μg/ml; Sigma). Retinal cells were cultured at 37 °C and 5% $CO_2$.

After 4 days in culture, retinal cells were fixed with 4% paraformaldehyde (PFA; Sigma). Cells were stained with primary antibodies against βIII-tubulin (1:2000; BioLegend, RRID:AB_2313773), detyrosinated tubulin (1:2000; Millipore), and phosphorylated (Tyr705) STAT3 (1:200; Cell Signaling Technology, RRID:AB_2491009). Secondary antibodies included donkey-anti-mouse and anti-rabbit antibodies conjugated to Alexa Fluor 488 or 594 (1:1000; ThermoFisher).

For studying neurite growth, all RGC neurites were photographed under a fluorescent microscope (200 x; Axio Observer.D1, Zeiss), and neurite length was determined using ImageJ software. To evaluate microtubule detyrosination, at least 30 axonal tips per well were photographed (400 x; Axio Observer.D1, Zeiss) and analyzed using ImageJ software. Axon tips were defined as the last 15 μm of βIII-tubulin positive neurites. For the quantification of STAT3 phosphorylation, at least 30 RGCs per well were evaluated (400 x; Axio Observer.D1, Zeiss).

## Quantification of pCRMP2 staining intensity in murine RGC cultures

To quantify pCRMP2 immunofluorescence, retinal cells were cultured in either 0.5 nM parthenolide or the vehicle. After 4 d in culture, cells were fixed and immunostained against βIII-tubulin (1:1,000; BioLegend; RRID:AB_2313773) and phospho(T514)-CRMP2 (1:2,000; Abcam; RRID:AB_942229). The staining intensity of pCRMP2 in βIII-tubulin–positive RGCs were analyzed using ImageJ software and calculated according to the following formula: Corrected total cell fluorescence = integrated density − (area of selected cell ×mean fluorescence of background reading). Data represent means ± SEM of three independent experiments. In each experiment, 80 RGCs per group were analyzed in two replicate wells.

## Dissociated human retinal cell cultures

Primary human retinal cultures were generated in two independent experiments with retinae isolated from two adult human patients' enucleated eyes.

The retinae were dissected from the eyecups, and the vitreous body was removed. Afterward, the retinae were cut into 8 pieces and digested separately, each in 10 ml Dulbecco's Modified Eagle medium (DMEM; Thermo Fisher) containing papain (10 U/ml; Worthington) and L-cysteine (0.2 μg/ml; Sigma) at 37 °C for 45 min. After digestion, retinae were washed with DMEM, triturated, and centrifuged in 50 ml DMEM (900 g, 5 min). Cell pellets were resuspended in a medium containing B27 supplement (1:50; Invitrogen) and penicillin/streptomycin (1:50; Merck) and passed through a cell strainer (40 μm; Greiner Bio-One). Retinal cultures were incubated with parthenolide (Sigma; 0.25–5.0 nM in dimethyl sulfoxide) or the vehicle. Some groups received recombinant rat CNTF (Peprotech; 200 ng/ml), either solely or combined with parthenolide. Cells were cultured at 37 °C and 5% $CO_2$ on 4-well plates (Nunc) (300 μl/well; 4 wells/group), which were coated with poly-D-lysine (0.1 mg/ml, molecular weight 70,000–150,000 Da; Sigma). After 4 days in culture, retinal cells were fixed with 4% paraformaldehyde (PFA; Sigma) and stained with primary antibodies against βIII-tubulin (1:2000;

BioLegend; RRID:AB_2313773), detyrosinated tubulin (1:2000; Millipore), and GAP43 (custom-made antibody, 1:1000; Invitrogen). Secondary antibodies included donkey-anti-mouse, and anti-rabbit antibodies conjugated to Alexa Fluor 488 or 594 (1:1000; ThermoFisher). Cultures were analyzed in the same way as described above for murine RGCs to study neurite growth and the percentage of detyrosinated axon tips.

### Intraperitoneal injections of DMAPT

Due to its high oral bioavailability of ~70% (*Guzman et al., 2007*) and the ability to cross the blood-brain barrier (*Hexum et al., 2015*), DMAPT can be delivered to the CNS via a systemic route of administration. Mice were given daily intraperitoneal injections of either DMAPT (0.2, 2, or 20 µg/kg; Abcam; dissolved in DMSO and further diluted in sterile PBS) or vehicle (DMSO diluted in sterile PBS). All spinal cord regeneration experiments were performed using the 2 µg/kg dosage. Treatment started directly after the optic nerve or spinal cord crush and was continued until the day of perfusion. Intraperitoneal injections were performed under blinded conditions.

### Basso Mouse Scale (BMS)

We scored mice using the Basso Mouse Scale to assess locomotor behavior after spinal cord injury (*Basso et al., 2006*). To this end, each mouse was placed separately into a round open field of 1 m in diameter and observed by two testers for 4 minutes. Scoring was based on different parameters, including ankle movement, paw placement, stepping pattern, coordination, trunk instability, and tail position, with a minimum score of 0 (no movement) to a maximum of 9 (normal locomotion). Before testing, mice were acclimated to the open field environment. BMS tests were performed before the injury or 1, 3, and 7 days after the injury, and then weekly. As quality criteria for lesion completeness, we used a BMS score of 0 on day 1 after injury and the absence of BDA or serotonin staining in transverse spinal cord sections (as described below). Mice that did not meet these criteria were excluded from the experiment. BMS tests were performed under blinded conditions.

### Immunohistochemistry

Animals were anesthetized and transcardially perfused with cold PBS, followed by 4% PFA in PBS. Brains, spinal cords, and eyes with attached optic nerves were isolated, post-fixed overnight in 4% PFA at 4 °C, and subsequently transferred to 30% sucrose for at least 4 hr for eyes and optic nerves or 5 days for brains and spinal cords.

Tissues were embedded in the KP-cryo compound (Klinipath) and frozen at –20 °C (retina, optic nerve, spinal cord) or on dry ice (brain). Sections were cut using a CM3050 S cryostat (Leica Biosystems). Optic nerves were cut into longitudinal sections (14 µm). For spinal cords of mice subjected to T8 crush, a segment from 3 mm rostral to 8 mm caudal from the injury site was cut into sagittal sections (20 µm). Spinal cord tissue, just rostral and caudal from this segment, was used for transverse sections (20 µm) to evaluate lesion completeness by identifying potentially spared axons. Tissue from the medullary pyramids ~1 mm above the pyramidal decussation was cut into coronal sections (20 µm). Sections were thaw-mounted onto charged glass slides (Superfrost Plus, VWR) and stored at –20 °C until further use.

The retinal, spinal cord, and brain sections were thawed, washed with PBS, and permeabilized for 10 min in methanol (Sigma). Sections were stained with primary antibodies against βIII-tubulin (1:1000; BioLegend, RRID: AB_10063408), phosphorylated STAT3 (1:200; Cell Signaling Technology, RRID:AB_2491009), serotonin (1:5000; Immunostar, RRID:AB_572262, RRID:AB_572263), GFP (1:500; Novus Biologicals, RRID:AB_10128178), NeuN (1:500; Abcam, RRID:AB_2532109), GFAP (1:500; Abcam, RRID:AB_305808), or CS-56 (1:70,000; Sigma, RRID:AB_476879). Secondary antibodies included donkey-anti-rabbit, anti-goat, and anti-mouse antibodies conjugated to Alexa Fluor 488, 594 (1:1000; ThermoFisher), or 405 (1:500; Jackson ImmunoResearch).

Retinal wholemounts were prepared without prior perfusion, fixed in 4% PFA for 30 min, and permeabilized in 2% Triton X-100 in PBS for 1 h. RGCs were stained using a primary antibody against βIII-tubulin (1:1000; BioLegend) and an anti-mouse secondary antibody conjugated to Alexa Fluor 488 (1:1000; ThermoFisher).

Sections and wholemounts were coverslipped with Mowiol (Sigma), and pictures were taken using a slide scanner (VS120; Olympus) or a confocal laser scanning microscope (SP8; Leica). Tissue analysis was performed under blinded conditions.

## Sample-size estimation, randomization and blinding procedure for *in vivo* experiments

For each experiment, a power analysis was performed beforehand to estimate the appropriate sample size. The power analysis was computed using the G*Power 3.1.7 software. Before the start of the experiment, individual vials containing DMAPT or vehicle (DMSO) stock solution were prepared for each experimental animal. The vials were randomized by a person who was neither involved in the implementation nor evaluated the experiments. These numbers were randomly distributed to mice of the same age and sex in different cages. This was carried out independently by another person who was neither involved in the data evaluation nor the randomization of the samples. This was followed by the execution of the experiments and the evaluation by scientists who were not involved in any randomization processes and did not know the identity of the injected samples. After completion of the data collection, values from mice with signs of spared axons were first removed from the data set for quality assurance. The criteria for this were a BMS Sore of a maximum of 0–1 on the first day after the lesion and the absence of uninjured serotonergic axons in spinal cord cross-sections >9–10 mm distal to the lesion site. Finally, the data points were assigned to the respective experimental groups by the person who initially blinded the vials.

## Quantification of regenerated axons in the optic nerve

Pictures of 6 longitudinal sections per animal were taken using a slide scanner (VS120; Olympus). The number of CTB-labelled axons extending 0.25, 0.5, and 0.75 mm (for DMAPT experiments) or 0.5, 1, 1.5, 2, and 2.5 mm beyond the lesion site (for hIL-6 +DMAPT experiments) was quantified and normalized to the width of the optic nerve at the respective measuring point. Each experimental group included 6–7 mice. Eyes with lens injury were excluded from the experiment.

## Quantification of RGCs in retinal wholemounts

Retinal wholemounts were divided into four quadrants. In each quadrant, 4 non-overlapping pictures were taken using a fluorescent microscope (400 x; Axio Observer.D1, Zeiss), proceeding from the center to the periphery. The number of βIII-tubulin positive RGCs was determined and normalized to an area of 1 $mm^2$. Values were averaged per retina and across all animals. Each experimental group included 5–9 retinae. Eyes with lens injury were excluded from the experiment.

## Quantification of RpST axon regeneration

Sagittal and coronal spinal cord sections were stained against serotonin and imaged with a slide scanner (20 x; VS120; Olympus). A grid with lines spaced every 500 μm was aligned to each image to count the number of serotonin-positive RpST axons at 500 μm intervals from 0 to 8 mm caudal from the lesion site. In addition, every other spinal cord section was analyzed, and the number of serotonin-positive axons was extrapolated to the total number of sections per spinal cord. Animals with spared serotonin-positive axons in transverse spinal cord sections >8 mm distal from the lesion site were excluded from the experiment. Each experimental group included 7–10 animals.

## Quantification of spinal cord lesion sizes and CSPG staining

Coronal spinal cord sections containing the T8 lesion site were stained against GFAP, and pictures were taken using a fluorescence microscope (Olympus VS120) to evaluate the lesion size after SCC. The lesion site, defined by the GFAP-positive astrocyte scar border, was traced and measured using ImageJ to obtain the total area. Furthermore, the width at the lesion center and both lateral edges were measured and averaged to determine the lesion width. The same sections were co-stained against CS-56 to visualize CSPGs at the injury site. Using ImageJ, the total area occupied by CSPGs was measured by tracing the CS-56 staining. Lesion sizes and CSPG staining was analyzed in three sections per animal.

## Statistics

Significances of intergroup differences were evaluated using Student's t-test or analysis of variance (ANOVA) followed by Tukey or Holm-Sidak post hoc tests using the Sigma STAT3.1 software (Systat Software).

## Acknowledgements

We thank Dr. Günter Gisselmann, Anastasia Andreadaki, Christopher Brennsohn, Davina Stoesser, and Kessy Brzozowski for their technical support. The German Research Foundation supported this work.

## Additional information

### Funding

| Funder | Grant reference number | Author |
|---|---|---|
| Bundesministerium für Bildung und Forschung | 03VP05200 | Dietmar Fischer |

The funders had no role in study design, data collection and interpretation, or the decision to submit the work for publication.

### Author contributions

Marco Leibinger, Conceptualization, Data curation, Formal analysis, Validation, Investigation, Writing – original draft, Writing – review and editing; Charlotte Zeitler, Formal analysis, Validation, Investigation, Writing – original draft, Writing – review and editing; Miriam Paulat, Philipp Gobrecht, Alexander Hilla, Anastasia Andreadaki, Investigation; Rainer Guthoff, Resources; Dietmar Fischer, Conceptualization, Data curation, Formal analysis, Supervision, Validation, Investigation, Writing – original draft, Project administration, Writing – review and editing

### Author ORCIDs

Marco Leibinger ![ORCID] http://orcid.org/0000-0001-6618-324X
Charlotte Zeitler ![ORCID] http://orcid.org/0000-0003-2318-2890
Dietmar Fischer ![ORCID] http://orcid.org/0000-0002-1816-3014

### Ethics

The use of human eyes and publishing of the obtained results was approved by the ethics committee of Heinrich Heine University Düsseldorf (study number 4067) and performed by the ethical standards as laid down in the 1964 Declaration of Helsinki and its later amendments or comparable ethical standards. (study number 4067). Informed consent, and consent to publish, was obtained.

This study was performed in strict accordance with the recommendations in the Guide for the Care and Use of Laboratory Animals of LANUV (Recklinghausen). All of the animals were handled according to approved institutional animal care and use committee protocols of the University Hospital Cologne. The protocol was approved by the Committee on the Ethics of Animal Experiments (LANUV). All surgery was performed under anesthesia, and every effort was made to minimize suffering. 84-02.04.2017. A218.

Reviewer #1 (Public Review): https://doi.org/10.7554/eLife.88279.3.sa1
Reviewer #2 (Public Review): https://doi.org/10.7554/eLife.88279.3.sa2
Reviewer #3 (Public Review): https://doi.org/10.7554/eLife.88279.3.sa3
Author Response https://doi.org/10.7554/eLife.88279.3.sa4

## Additional files

### Supplementary files
• MDAR checklist

### Data availability
All data of the paper are available in the manuscript.

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
