## [Editor Report · eLife assessment]

The primary goal of this paper is to examine microtubule detyrosination as a potential therapeutic target for axon regeneration. The **valuable** findings of this study provide **convincing** evidence for mechanistic links between microtubule detyrosination and neurite outgrowth in vitro and some evidence for axon regeneration in vivo.

---

## [Referee Report · Reviewer #1 (Public Review)]

This manuscript by Leibinger et al describes their results from testing an interesting hypothesis that microtubule detyrosination inhibits axon regeneration and its inhibitor parthenolide could facilitate axon regeneration and perhaps functional recovery. Overall, the results from in vitro studies are largely well performed. However, the in vivo data are less convincing.

Interpretation of the findings in this study are limited by several gaps:

1. It is unclear whether microtubule detyrosination a primary effect of hIL-6 and PTEN deletion or secondary to the increased axon growth?

2. Is there any direct evidence for Akt and/or JAK/Stat3 to promote microtubule detyrosination?

3. What is the impact of parthenolide on cell soma of neurons and other cell types?

4. Direct evidence that parthenolide augments PTEN deletion in optic nerve or spinal cord is not provided.

5. Serotonergic neurotoxin DHT ablates both regenerating and non-regenerating serotonergic axons, which makes spinal cord findings it difficult to interpret.

6. DMAPT was given by i.p. injection. What happens to microtubule detyrosination in other cells within and outside of CNS?

---

## [Referee Report · Reviewer #2 (Public Review)]

In the current study, Fischer and colleagues extensively examined the role of parthenolide in inhibiting microtubule detyrosination and making the mechanistic link for the compound to facilitate the role of IL6 and PTEN/KO in promoting neurite outgrowth and axon regeneration. The in vitro and mechanistic work laid the foundation for the authors to reach several key predictions that such detyrosination can be applied for in vivo applications. Thus the authors extended the work to optic nerve regeneration and spinal cord recovery. The in vivo compound that the authors utilized is DMAPT, which plays a synergistic role with existing pro-regeneration therapies, such as Il6 treatment.

The major strength of the work is the first half of the mechanistic inquiries, where the authors combined cell biology and biochemistry approaches to dissect the mechanistic link from parthenolide to microtube dynamics. The shortcoming is that the in vivo data is limited, and the effects might be considered mild, especially by benchmarking with other established and effective strategies.

The work is solid and prepares a basis for others to test the role of DMAPT in other settings, especially in the setting of other effective pro-regenerative approaches. With the goal of comprehensive and functional recovery in vivo, the impact of the work and the utilities of the methods remain to be tested broadly in other models in vivo.

---

## [Referee Report · Reviewer #3 (Public Review)]

The primary goal of this paper is to examine microtubule detyrosination as a potential therapeutic target for axon regeneration. Using dimethylamino-parthenolide (DMAPT), this study extensively examines mechanistic links between microtubule detyrosination, interleukin-6 (IL-6), and PTEN in neurite outgrowth in retinal ganglion cells in vitro. These findings provide convincing evidence that parthenolide has a synergistic effect on IL-6- and PTEN-related mechanisms of neurite outgrowth in vitro. The potential efficacy of systemic DMAPT treatment to promote axon regeneration in mouse models of optic nerve crush and spinal cord injury was also examined.

Strengths

1. The examination of synergistic activities between parthenolide, hyperIL-6, and PTEN knockout is leveraged not only for potential therapeutic value, but also to validate and delineate mechanism of action.

2. The in vitro studies, including primary human retinal ganglion cells, utilize a multi-level approach to dissect the mechanistic link from parthenolide to microtubule dynamics.

3. The studies provide a basis for others to test the role of DMAPT in other settings, particularly in the context of other effective pro-regenerative approaches.

Weaknesses

1. In vivo studies are limited to select outcomes of recovery and do not validate or address mechanism of action in vivo.

---

## [Author Response]

The following is the authors’ response to the current reviews.

**Reviewer #1 (Public Review):**
This manuscript by Leibinger et al describes their results from testing an interesting hypothesis that microtubule detyrosination inhibits axon regeneration and its inhibitor parthenolide could facilitate axon regeneration and perhaps functional recovery. Overall, the results from in vitro studies are largely well performed. However, the in vivo data are less convincing.Interpretation of the findings in this study are limited by several gaps:1. It is unclear whether microtubule detyrosination a primary effect of hIL-6 and PTEN deletion or secondary to the increased axon growth?

This point is based on a misunderstanding, as shown in Fig. 2 by Western blot, that detyrosination was increased after intravitreal injection of AAV2-hIL-6 into optic nerves. These optic nerves were uninjured! This indicates that the increased detyrosination is an effect of the treatment itself and does not occur due to axonal regeneration.

Why hIL-6 and PTEN nevertheless increase axonal regeneration is because the positive effect on other signaling pathways, such as JAK/STAT3 and mTOR, ultimately predominates. Consequently, we show, for both PTEN ko and hIL-6, that we can further enhance these positive effects by neutralizing the negative aspect of increased detyrosination using DMAPT.

1. Is there any direct evidence for Akt and/or JAK/Stat3 to promote microtubule detyrosination?

Regarding the AKT/GSK3 signaling pathway, it has been well described that GSK3 activity leads to phosphorylation of microtubule-associated protein 1B, which results in enhanced tubulin detyrosination (Lucas et al., 1998, Goold et al 1999, Owen and Gordon-Weeks 2003). As shown in our previous and cited work, hIL-6 promotes the activation of AKT, which in turn inhibits GSK3 (Leibinger et al. 2016). In Fig. 2, we have also shown that intravitreal hIL-6 treatment in the optic nerve leads to increased inhibitory phosphorylation of GSK3 at the target site of AKT, and that tubulin detyrosination is increased. The same was also shown for PTEN ko: In a previous publication, we showed that PTEN ko increases AKT activity, inhibiting GSK3 phosphorylation (Leibinger et al. 2019). In Fig. 3 of the actual study, we show that PTEN ko results in enhanced tubulin detyrosination. In conclusion, treatments activating the AKT/GSK3 signaling enhance tubulin detyrosination.

On the other hand, JAK/STAT3 has no direct effect on detyrosination. This was demonstrated in experiments using the CNTF application, which reportedly activates the JAK/STAT3 pathway without affecting AKT/GSK3 (Leibinger et al, 2009, 2016, 2017).

In cell culture, we have shown that activation of the JAK/STAT3 pathway by CNTF does not change tubulin detyrosination in neurites (Fig. 1 H, I, M; N). Moreover, DMAPT in RGC’s cell bodies does not affect the phosphorylation of STAT3 and S6, and thus has no measurable effect on JAK/STAT3 or the mTOR pathway.

1. What is the impact of parthenolide on cell soma of neurons and other cell types?

Parthenolide and DMAPT show a regenerative effect in the nanomolar range (cell culture) and a bell-shaped concentration-response curve. We show a close correlation between detyrosinated microtubules and regeneration (with and without hIL-6 or PTEN-KO), which is, in our opinion, convincing. Moreover, we would like to address a likely misunderstanding in this comment and provide further clarification. The detyrosination of alpha-tubulin occurs after its attachment to microtubules through the action of the tubulin carboxy peptidase vasohibin 1 and 2 (Vash 1, 2). Consequently, tubulin is already present in the detyrosinated form within existing microtubules, and the administration of DMAPT does not affect these pre-existing microtubules. However, DMAPT does play a crucial role in preventing the detyrosination of newly attached tubulin dimers in the growth cones of developing axons. This explains why we can detect detyrosinated tubulin specifically in those regions and why our immunohistochemical analyses in the cell culture experiments focused solely on axon tips.

It is important to note that when used at low concentrations, which promote axon growth, DMAPT does not measurably affect detyrosination in other neuronal compartments, such as the RGCs' somata. We might observe a decrease in detyrosination only at much higher concentrations. However, this outcome would be inconsequential to our findings.

Whether additional effects of DMAPT contribute to improved regeneration is not excluded, although unlikely. If so, their investigation would be beyond the scope of the current paper.

1. Direct evidence that parthenolide augments PTEN deletion in optic nerve or spinal cord is not provided.

Our research paper primarily investigates the combination of DMAPT with h-IL-6. We chose to combine DMAPT with hIL-6 because, unlike PTEN-KO, only hIL-6 has been demonstrated to facilitate functional recovery following a complete spinal cord crush injury (Leibinger et al., 2021). Therefore, it is unclear why conducting in vivo experiments with PTEN-KO would be necessary, which cannot be used therapeutically. Since we have shown the beneficial effects of DMAPT on hIL-6 in two different in vivo models (optic nerve and spinal cord) anatomically and functionally, we feel that the repetition of these experiments with PTEN ko, which has no therapeutic implication, would not justify the sacrifice of additional animals. This would contradict the principles of reduction, refinement, and replacement, aiming to minimize the use of animals in our research.

In contrast, the PTEN experiments primarily serve to support the underlying mechanism and demonstrate that DMAPT generally counteracts the negative effect on MT detyrosination, even in conjunction with other procedures that activate the PI3K/AKT pathway. These findings were mechanistically elucidated through cell culture experiments utilizing immunohistochemial analysis, which the editors highlighted as strengths of our paper.

1. Serotonergic neurotoxin DHT ablates both regenerating and non-regenerating serotonergic axons, which makes spinal cord findings it difficult to interpret.

The impact of unregenerated serotonergic axons on stereotypic hind leg movements, as assessed through BMS analysis, appears to be minimal, as demonstrated in our previous study (Leibinger et al., 2021). Specifically, our findings revealed that depleting serotonergic neurons using DHT did not significantly affect the BMS score in uninjured animals (Leibinger et al., 2021). Furthermore, even in the control group comprising animals with spinal cord lesions where anatomical regeneration of the RpST did not occur, the administration of DHT had no discernible effect (Fig. 7 K, L).

To address this concern, we included the following information in the revised manuscript: "It might be considered plausible that the depletion of non-regenerated serotonergic axons could have contributed to these results. However, we can largely dismiss this possibility, as DHT did not influence the non-regenerated vehicle control group. Additionally, in a previous publication, we have demonstrated that the general depletion of serotonergic neurons in uninjured animals also does not significantly impact open field locomotion, as measured by the BMS score and subscore (Leibinger et al., 2021)."

1. DMAPT was given by i.p. injection. What happens to microtubule detyrosination in other cells within and outside of CNS?

This question is the same as raised under point 3. -> response see 3.

**Reviewer #2 (Public Review):**
In the current study, Fischer and colleagues extensively examined the role of parthenolide in inhibiting microtubule detyrosination and making the mechanistic link for the compound to facilitate the role of IL6 and PTEN/KO in promoting neurite outgrowth and axon regeneration. The in vitro and mechanistic work laid the foundation for the authors to reach several key predictions that such detyrosination can be applied for in vivo applications. Thus the authors extended the work to optic nerve regeneration and spinal cord recovery. The in vivo compound that the authors utilized is DMAPT, which plays a synergistic role with existing pro-regeneration therapies, such as Il6 treatment.The major strength of the work is the first half of the mechanistic inquiries, where the authors combined cell biology and biochemistry approaches to dissect the mechanistic link from parthenolide to microtube dynamics. The shortcoming is that the in vivo data is limited, and the effects might be considered mild, especially by benchmarking with other established and effective strategies.The work is solid and prepares a basis for others to test the role of DMAPT in other settings, especially in the setting of other effective pro-regenerative approaches. With the goal of comprehensive and functional recovery in vivo, the impact of the work and the utilities of the methods remain to be tested broadly in other models in vivo.
**Reviewer #3 (Public Review):**
The primary goal of this paper is to examine microtubule detyrosination as a potential therapeutic target for axon regeneration. Using dimethylamino-parthenolide (DMAPT), this study extensively examines mechanistic links between microtubule detyrosination, interleukin-6 (IL-6), and PTEN in neurite outgrowth in retinal ganglion cells in vitro. These findings provide convincing evidence that parthenolide has a synergistic effect on IL-6- and PTEN-related mechanisms of neurite outgrowth in vitro. The potential efficacy of systemic DMAPT treatment to promote axon regeneration in mouse models of optic nerve crush and spinal cord injury was also examined.Strengths1. The examination of synergistic activities between parthenolide, hyperIL-6, and PTEN knockout is leveraged not only for potential therapeutic value, but also to validate and delineate mechanism of action.1. The in vitro studies, including primary human retinal ganglion cells, utilize a multi-level approach to dissect the mechanistic link from parthenolide to microtubule dynamics.1. The studies provide a basis for others to test the role of DMAPT in other settings, particularly in the context of other effective pro-regenerative approaches.Weaknesses1. In vivo studies are limited to select outcomes of recovery and do not validate or address mechanism of action in vivo.
**Reviewer #1 (Recommendations For The Authors):**
Overall, it doesn't seem like the authors bought into or addressed any issues raised during the previous review. In testing their central hypothesis, a critical experiment was to assess the outcome of PTEN knockout in combination with their novel treatment (parthenolide or DMAPT). Unfortunately, this and other issues have not been addressed in this revision.

PTEN is not part of our central hypothesis. Our research paper primarily investigates the combination of DMAPT with h-IL-6. We chose to combine DMAPT with hIL-6 because, unlike PTEN-KO, only hIL-6 has been demonstrated to facilitate functional recovery following a complete spinal cord crush injury (Leibinger et al., 2021). Therefore, it is unclear why conducting in vivo experiments with PTEN-KO would be necessary, which cannot be used therapeutically. Since we have shown the beneficial effects of DMAPT on hIL-6 in two different in vivo models (optic nerve and spinal cord) anatomically and functionally, we feel that the repetition of these experiments with PTEN ko, which has no therapeutic implication, would not justify the sacrifice of additional animals. This would contradict the principles of reduction, refinement, and replacement, aiming to minimize the use of animals in our research.

In contrast, the PTEN experiments primarily serve to support the underlying mechanism and demonstrate that DMAPT generally counteracts the negative effect on MT detyrosination, even in conjunction with other procedures that activate the PI3K/AKT pathway. These findings were mechanistically elucidated through cell culture experiments utilizing immunohistochemial analysis, which the editors highlighted as strengths of our paper.

**Reviewer #2 (Recommendations For The Authors):**
The response and revision provided here did not improve the manuscript - the authors chose to focus on re-organizing the methods but did not provide any new experimental data. Thus my recommendations remain the same as the previous round. In brief, the in vivo evidence was rather weak, especially if no further evidence was offered to respond to these points below.To possibly improve the manuscript, the authors could consider enhancing the in vivo parts in the following manner;1. possibly detyrosination staining in the optic nerve vertical section - it would be interesting to see how the detyrosination assays may work for regenerating conditions, or as an alternate, the authors may consider retina tissue biochemistry (with & without IL6, with & without DMAPT) repeating the biochemical assays as established Fig 2B –

The detyrosination of alpha-tubulin occurs after its attachment to microtubules through the action of the tubulin carboxy peptidase vasohibin 1 and 2 (Vash 1, 2). Consequently, tubulin is already present in the detyrosinated form within existing microtubules, and the administration of DMAPT does not affect these pre-existing microtubules. However, DMAPT does play a crucial role in preventing the detyrosination of newly attached tubulin dimers in the growth cones of developing axons. This explains why we can detect detyrosinated tubulin specifically in those regions and why our immunohistochemical analyses in the cell culture experiments focused solely on axon tips.

It is important to note that when used at low concentrations, which promote axon growth, DMAPT does not measurably affect detyrosination in other neuronal compartments, such as the RGCs' somata. We might observe a decrease in detyrosination only at much higher concentrations. Because of these reasons, we could not clearly identify and stain axon tips in 14 µm thick optic nerve sections.

1. How do the authors benchmark the DMAPT retreatment in the setting of PTEN (aav2-cre injection for cKO) and /or PTEN/SOCS3/CNTF dKO? Which are the best approaches to promote optic nerve regeneration? Would the authors expect DMAPT retreatment to be synergetic with PTENcKO?

Based on our previous findings, we anticipate that DMAPT would exhibit a synergistic effect when combined with PTEN ko, as demonstrated in our in vitro studies with cultured neurons. Additionally, synergistic effects between DMAPT and PTEN/SOCS3 dKO +CNTF are possible. While these hypotheses hold promise, our current paper primarily focuses on combining DMAPT with hIL-6, which has consistently shown remarkable efficacy as a standalone treatment in optic nerve regeneration.

1. Regarding the DMAPT treatment, one notable issue was that the RGC survival subject to ONC was very poor, which may limit the effects of DMAPT daily injection. The authors may consider further combining DMAPT with the DLK/LZK inhibitors to examine the synergistic effects.

As DMAPT itself is not neuroprotective and does not affect retinal ganglion cells' (RGCs) regenerative state by inducing the expression of regeneration-associated genes, a combination with a neuroprotective and regenerative treatment would show stronger effects. This is exactly what we found when combining DMAPT with neuroprotective hIL-6 (Leibinger et al. 2016) in the current paper.

Moreover, in the raphespinal tract, where respective neurons do not undergo apoptotic cell death after axotomy, the DMAPT effect on anatomic axon regeneration was stronger than in the optic nerve, even without combination with hIL-6, with some axons reaching distances of up to 7 mm distal to the lesion. So, DMAPT can induce long-distance regeneration in neuronal populations unaffected by cell death. Therefore, additional experiments with DLK/LZK inhibitors, as suggested by this reviewer, would not provide an additional benefit to our paper and would not justify the additional sacrifice of animal lives.

1. Overall, the phenotypes in Figs 5-8 were rather weak after DMAPT treatment, which are universal challenges to spinal cord regeneration. The authors may present this section of the data with further clarification on the selection standards in the methods, such as how the animals and treatment were selected and how a double-blinded experimental design may help further evaluate the effects of DMAPT treatment. I found little relevant information in the current manuscript.

In the anatomic and functional regeneration analysis presented in Fig. 5-8, we only included animals with a BMS score of 0 one day after the spinal cord crush, indicating a complete absence of hind leg movement. Furthermore, we employed immunohistochemical staining to ensure that no serotonergic axons were detected at 8-10 mm from the lesion site in any of the animals, thus confirming the thoroughness of the lesion (Supplementary Fig. 4). Both the evaluation of the BMS score and the assessment of anatomical regeneration was conducted in a double-blinded manner, ensuring unbiased and objective observations. To address this concern, we will add the following paragraph in the M&M part:

“Blinding procedure for in vivo experimentsBefore the start of the experiment, individual vials containing DMAPT or vehicle (DMSO) stock solution were prepared for each particular experimental animal. The vials were randomized by a person who was neither involved in the implementation nor in the evaluation of the experiments. These numbers were randomly distributed to mice of the same age and sex in different cages. This was carried out independently by another person who was neither involved in the data evaluation nor the randomization of the samples. This was followed by the execution of the experiments and the evaluation by scientists who were not involved in any of the randomization processes and did not know the identity of the injected samples. After completion of the data collection, values from mice with signs of spared axons were first removed from the data set for reasons of quality assurance. The criteria for this were a BMS-Sore of a maximum of 0-1 on the first day after the lesion and the absence of uninjured serotonergic axons in spinal cord cross-sections >8-10 mm distal to the lesion site. Finally, the data points were assigned to the respective experimental groups by the person who initially blinded the vials.”

**Reviewer #3 (Recommendations For The Authors):**
Addition of supporting data, revision of discussion, and inclusion of references for parthenolide activities improved the manuscript and adequately addressed concerns

The following is the authors’ response to the original reviews.

We feel that the use of human RGCs should be considered a highlight and strength of our paper because, as far as we know, our study is the first to utilize human primary cultures of RGCs to confirm the effectiveness of drugs on human cells. Therefore, this might be of interest to colleagues in our field. Moreover, we have added additional data as suppl. Fig. proving that these cells are living RGCs so this concern has been addressed. In addition, we provide further explanations why other activities of DMAPT beyond microtubule detyrosination, such as oxidative stress and NFkB inhibition, are not considered in experimental examinations or in the interpretation of findings. Therefore, we strongly recommend that this point should not be considered a weakness.

Strengths:1. The examination of synergistic activities between parthenolide, hyper-IL-6, and PTEN knockout is leveraged not only for potential therapeutic value, but also to validate and delineate mechanism of action.1. The in vitro studies utilize a multi-level approach that combines cell biology and biochemistry approaches to dissect the mechanistic link from parthenolide to microtubule dynamics.1. The studies provide a basis for others to test the role of DMAPT in other settings, particularly in the context of other effective pro-regenerative approaches.Weaknesses:1. In vivo studies are limited to select outcomes of recovery and do not validate or address mechanism of action in vivo.1. Known activities of DMAPT beyond microtubule detyrosination, such as oxidative stress, mitochondrial function and NFkB inhibition, are not considered in experimental examinations or in the interpretation of findings.

Our research indicates that parthenolide exhibits a regenerative effect within a nanomolar range and with a bell-shaped concentration-response curve in culture. Moreover, we demonstrate a close correlation between the inhibition of detyrosinated microtubules and regeneration and consider the effects of hIL-6 or PTEN-KO on detyrosination in mouse and human RGCs. Therefore, we offer a coherent and satisfactory mechanistic explanation for the effects of parthenolide. We, therefore, feel the request to experimentally explore additional, somewhat speculative possibilities is not reasonable or helpful, and this issue should not be considered as a weakness. Moreover, to the best of our knowledge, no evidence suggests profound antioxidative effects of DMAPT or parthenolide within these low-concentration ranges and that these would affect axon regeneration. Antioxidative effects may also not explain the observed bell-shaped curve. Furthermore, we have already considered the effect of NFkappaB in our previous work (Gobrecht et al., 2016) and shown that NFkappaB remains unaffected by low concentrations of parthenolide. Hence, conducting additional experiments addressing oxidative stress or other speculative causes will not strengthen our findings and do not justify the additional sacrifice of animal lives.

Nevertheless, we added the following sentence in our manuscript to address this issue: “Although we cannot exclude the possibility that other known activities of parthenolide/DMAPT, such as oxidative stress or NF-kB inhibition, could have contributed to the observed effects, this is rather unlikely because such effects have only been reported at much higher micromolar concentrations (Bork et al., 1997; Saadane et al., 2007; Carlisi et al., 2016; Gobrecht et al., 2016).”

Editorial Comments:The reviewers' consensus is that this manuscript, although containing an impressive amount of data, lacks cohesion.The mechanistic studies in vitro are of a distinctly different caliber than the in vivo studies. Additional data is needed to demonstrate that the mechanisms delineated in vitro are related to the outcomes in vivo. As is, this reads as a comprehensive in vitro study with premature in vivo data tacked on the end.The manuscript should contain the necessary background and contextual information needed to fully understand the work. Clarity of rationale and context for experimental method/design (why one reagent or insult is selected over another), result interpretation (what does this data tell you and not tell you), and implications for results (what does this mean in the context of current knowledge) should be improved throughout.Technical:1. There is no validation of human RGC cultures. If this data is to remain in the manuscript, proper verification data should be provided to demonstrate that these are indeed RGCs and that they are viable.

The retinal ganglion cells (RGCs) were identified by applying the same criteria as murine and rat RGCs,encompassing morphological and immunohistochemical criteria. The staining of a piece of human retina (see Author response image 1) shows βIII-tubulin-positive cells in the ganglion cell layer and forming axonal bundles in the fiber layer. These are RGCs, and it is confirmed that the βIII-tubulin antibody stains human RGCs (Author response image 1A). In addition, the somata of these human RGCs in the retina have a similar diameter (somewhat larger than murine RGCs Author response image 1A, B) to the cultured βIII-tubulin-positive cells (RGCs) and a similar morphology. Finally, these regenerating neurons are GAP43-positive, a regeneration-associated protein shown in Author response image 1C. Thus, these data prove that the cultured cells were human RGCs. These data were included as a suppl. Fig. 1.

The viability of the neurons was confirmed, as evidenced by their ability to grow neurites - a clear indication of their vitality. We also verified the viability by calceinstaining.

As far as we know, our study is the first to utilize human primary cultures of RGCs to confirm the effectiveness of CNTF and parthenolide on human cells. Therefore, we would have expected this accomplishment to be emphasized as a strength of our paper.

**Author response image 1. sa4fig1:** Evidence for viable, regenerating human retinal ganglion cells. (**A**) Retinal flat mounts from human (left) and mouse (right) stained for βIII-tubulin. Scale bar: 50 μm. (**B**) Human (left) and mouse (right) RGCs cultured for 4 days and stained for βIII-tubulin. Scale bar: 25 μm. (**C**) Human βIIItubulin-positive RGCs with regenerating neurites are also GAP43-positive. Scale bar: 50 μm

1. For graphs depicting means and errors, it is advised that the authors evaluate their use of SEM. Standard deviation should be used when illustrating the distribution of measurements/individuals within a population. Standard error should be used for determining accuracy of the calculated mean, i.e. how close are individuals to the calculated mean? Since standard error is a measure of accuracy rather than distribution, it moves towards zero as the population size increases, regardless of the distribution. Thus, error bars intended to show the range of an effect (i.e. how much functional recovery with treatment?), should be depicted as standard deviation, which illustrates the actual range of data.

To provide best possible transparency we incorporated each individual data point within our graphs, thus offering a detailed depiction of the complete range of effects. We firmly believe that this approach provides enhanced clarity compared to a standard deviation and grants a more comprehensive understanding of the data. It is worth noting that also presenting the standard error adds supplementary information regarding the accuracy of the calculated mean.

Thus, we firmly stand by our chosen method of data presentation, as we believe it furnishes readers with more valuable insights. However, if there are additional compelling arguments to display the standard deviation instead of the standard error, we are more than willing to consider them.

1. One notable issue was that the RGC survival subject to ONC was very poor, which maylimit the effects of DMAPT daily injection. The authors may consider further combining DMAPT with the DLK/LZK inhibitors to examine the synergistic effects.

As DMAPT itself is not neuroprotective and does not affect retinal ganglion cells' (RGCs) regenerative state by inducing the expression of regeneration-associated genes, a combination with a neuroprotective and regenerative treatment would show stronger effects. This is exactly what we found when combining DMAPT with neuroprotective hIL-6 (Leibinger et al. 2016) in the current paper.

Moreover, in the raphespinal tract, where respective neurons do not undergo apoptotic cell death after axotomy, the DMAPT effect on anatomic axon regeneration was stronger than in the optic nerve, even without combination with hIL-6, with some axons reaching distances of up to 7 mm distal to the lesion. So, DMAPT can induce long-distance regeneration in neuronal populations unaffected by cell death. Therefore, we feel that additional experiments with DLK/LZK inhibitors, as suggested by this reviewer, would not provide an additional benefit to our paper and not justify the additional sacrifice of animal lives.

To address this issue, we added the following paragraph: “Expectedly, DMAPT was not able to protect RGCs from axotomy-induced cell death (Fig. 4 F, G) since it does solely accelerate microtubule polymerization in axonal growth cones without affecting neuroprotective signaling pathways in the cell body (Fig. 1 F, G; supplementary Fig. 2). We then repeated these experiments in combination with intravitreally applied AAV2hIL-6 which reportedly has a significant neuroprotective effect (Leibinger et al., 2016) (Fig. 4 H).”

1. Serotonergic neurotoxin DHT, which in the spinal cord injury model ablates both regenerating and nonregenerating serotonergic axons, which makes interpretation of the results difficult. This should be addressed directly in interpretation and discussion.

The impact of unregenerated serotonergic axons on stereotypic hind leg movements, as assessed through BMS analysis, appears to be minimal, as demonstrated in our previous study (Leibinger et al., 2021). Specifically, our findings revealed that depleting serotonergic neurons using DHT did not significantly affect the BMS score in uninjured animals (Leibinger et al., 2021). Furthermore, even in the control group comprising animals with spinal cord lesions where anatomical regeneration of the RpST did not occur, the administration of DHT had no discernible effect (Fig. 7 K, L).

To address this concern, we propose including the following information in the revised manuscript: "It might appear conceivable that the depletion of non-regenerated serotonergic axons may have contributed to these results. However, we can rule this out since DHT did not influence the non-regenerated vehicle control group. Furthermore, we have shown in a previous publication that the general depletion of serotonergic neurons in uninjured animals also has no significant influence on openfield locomotion as measured in the BMS score and subscore (Leibinger et al., 2021). Furthermore, we have shown in a previous publication that the general depletion of serotonergic neurons in uninjured animals also has no significant influence on openfield locomotion as measured in the BMS score and subscore (Leibinger et al., 2021).”

5). Overall, the phenotypes in Figs 5-8 were rather weak after DMAPT treatment, which are universal challenges to spinal cord regeneration. The authors may present this section of the data with further clarification on the selection standards in the methods, such as how the animals and treatment were selected and how a double-blinded experimental design may help further evaluate the effects of DMAPT treatment. I found little relevant information in the current manuscript.

In the anatomic and functional regeneration analysis presented in Figures 5-8, we only included animals with a BMS score of 0 one day after the spinal cord crush, indicating a complete absence of hind leg movement. Furthermore, we employed immunohistochemical staining to ensure that no serotonergic axons were detected at 8-10 mm from the lesion site in any of the animals, thus confirming the thoroughness of the lesion (Supplementary Fig. 4). Both the evaluation of the BMS score and the assessment of anatomical regeneration was conducted in a doubleblinded manner, ensuring unbiased and objective observations. To address this concern, we will add the following paragraph in the M&M part:

“Blinding procedure for in vivo experimentsBefore the start of the experiment, individual vials containing DMAPT or vehicle (DMSO) stock solution were prepared for each experimental animal. The vials were randomized by a person who was neither involved in the implementation nor evaluated the experiments. These numbers were randomly distributed to mice of the same age and sex in different cages. This was carried out independently by another person who was neither involved in the data evaluation nor the randomization of the samples. This was followed by the execution of the experiments and the evaluation by scientists who were not involved in any randomization processes and did not know the identity of the injected samples. After completion of the data collection, values from mice with signs of spared axons were first removed from the data set for quality assurance. The criteria for this were a BMS Sore of a maximum of 0-1 on the first day after the lesion and the absence of uninjured serotonergic axons in spinal cord cross-sections >9-10 mm distal to the lesion site. Finally, the data points were assigned to the respective experimental groups by the person who initially blinded the vials.”

1. Several supplemental figures are discussed as critical elements of the studies performed. The authors are encouraged to include figures discussed as primary data as primary figures in the manuscript and provide the necessary information regarding experimental design and methods, including "n".

Thank you for the suggestion.

1. While the "n" is clear for some subsets of figures (as noted in the rebuttal), it is not clear for all outcomes/figure subsets. For example, it appears that some outcomes were performed in only a subset of the total experimental population and not in the context of statistically significant result. A good example of this is the figure for in vivo suboptimal dosing. The experimental design suggests n=7-10, but the group considered suboptimal due to statistical insignificance is listed as n=4. Is this an entirely separate cohort? If so, is n=4 sufficient and was it considered statistically in the context of the higher-powered cohorts? The lack of clarity regarding experimental design should be addressed.

To ensure transparency we have provided all n-numbers for each outcome and figure subset. Additionally, the precise n-numbers can be inferred by observing the number of individual points depicted in the graphs. All statistical data are appropriately indicated in the figure legends for reference.

The data presented in suppl. Fig. 3 represents a preliminary experiment to find effective doses of DMAPT in vivo. In this initial phase, we tested three different doses of DMAPT (0.2, 2, 20 µg/kg) in a reduced group size of only four animals per group. This reduction in animal numbers aligns with the principles to determine reduction, refinement, and replacement, aiming to minimize the use of animals in our research. Subsequently, the group demonstrating the most robust effect (2 µg/kg) was expanded by including additional animals to meet the a priori calculated sample size and validate the results. These additional animal data are presented in Figure 4 A-C.In the case of suppl. Fig. 3 A, B the statistical analysis indicated a significant effect in A using an n=4. As a result, there was no need to utilize additional animals for this particular experiment.

Gaps:1. By in vitro studies, the authors showed that hIL-6 treatment or PTEN knockout elevated microtubule detyrosination. But when does this occur? In another words, is this a primary effect of these treatments or secondary to the increased axon growth? How does this fit with the observations that these interventions promote axon regeneration both in vitro and in vivo?

This point also seems to be based on a misunderstanding, as shown in Figure 2 by Western blot, that detyrosination was increased after intravitreal injection of AAV2-hIL-6 into optic nerves. These optic nerves were uninjured! This indicates that the increased detyrosination is an effect of the treatment itself and does not occur due to axonal regeneration.

Why hIL-6 and PTEN nevertheless increase axonal regeneration is because the positive effect on other signaling pathways, such as JAK/STAT3 and mTOR, ultimately predominates. Consequently, we show, for both PTEN ko and hIL-6, that we can further enhance these positive effects by neutralizing the negative aspect of increased detyrosination using DMAPT.

1. Is there any direct evidence for Akt and/or JAK/Stat3 to promote microtubule detyrosination?

As described in our previous and cited work, hIL-6, in contrast to CNTF, promotes the activation of AKT (Leibinger et al. 2016). In Fig. 2, we have also shown that intravitreal hIL-6 treatment in the optic nerve leads to increased phosphorylation of GSK3, a substrate of AKT, and that tubulin detyrosination is increased.

As far as we know, JAK/STAT3 has no direct effect on detyrosination.

In cell culture, we have shown that activation of the JAK/STAT3 pathway by CNTF application does not change tubulin detyrosination in neurites (Fig. 1 H, I, M; N).

DMAPT in RGC’s cell bodies does not affect the phosphorylation of STAT3 and S6, and thus has no measurable effect on JAK/STAT3 or the mTOR pathway. Moreover, tubulin detyrosination in neuronal cell bodies is not affected by DMAPT.

1. Empirical data linking in vivo regeneration with mechanisms delineated in in vitro studies is limited. The addition of such data (i.e. biochemical assays, relevant histology) would better enable interpretation of in vivo studies and improve cohesiveness of the work as a whole.

The mechanistic links between hIL-6 /PTEN-signaling and tubulin detyrosination and the abrogation of the adverse effects by DMAPT have been extensively addressed in vitro, which has been positively highlighted here in several places. Indeed, the in vivo data were intended to mainly confirm that the mechanisms elaborated in vitro are relevant to axonal regeneration and functional restoration in vivo. Most importantly our data demonstrate that systemic DMAPT application promotes axon regeneration in the CNS and improves functional recovery after a complete spinal cord injury. Form a clinical point of view this is important.

1. DMAPT activities are not limited to microtubule detyrosination. These alternate activities should be considered, particularly in in vivo studies. Empirical evidence of the potential impact for these mechanisms in the retina, optic nerve, and systemically is strongly encouraged. In vitro studies or studies of a specific neuronal population are insufficient to extrapolate activities in an intact system.

Parthenolide and DMAPT show a regenerative effect in the nanomolar range (cell culture) and a bell-shaped concentration-response curve. We show a close correlation between detyrosinated microtubules and regeneration (with and without hIL6 or PTEN-KO), which is, in our opinion, convincing. Whether additional effects of DMAPT contribute to improved regeneration is not excluded, although unlikely. If so, their investigation would be beyond the scope of the current paper.

1. How do the authors benchmark the DMAPT retreatment in the setting of PTEN (aav2-cre injection for cKO) and /or PTEN/SOCS3/CNTF dKO? Which are the best approaches to promote optic nerve regeneration? Would the authors expect DMAPT retreatment to be synergetic with PTENcKO?

Based on our previous findings, we anticipate that DMAPT would exhibit a synergistic effect when combined with PTEN ko, as demonstrated in our in vitro studies with cultured neurons. Additionally, synergistic effects between DMAPT and PTEN/SOCS3 dKO +CNTF are possible. While these hypotheses hold promise, our current paper primarily focuses on combining DMAPT with hIL-6, which has consistently shown remarkable efficacy as a standalone treatment in optic nerve regeneration.

Furthermore, our rationale for combining DMAPT with hIL-6 rather than PTEN-KO stems from the fact that, unlike PTEN-KO, hIL-6 has been proven to enable functional recovery following complete spinal cord crush injuries (Leibinger et al., 2021).

1. A cohesive discussion of findings would be beneficial. What can and cannot be elucidated from in vitro and in vivo studies? How does the in vivo effect compare to existing strategies? What are the limitations of the studies performed? Are there alternative explanations for the findings in vitro or in vivo?

We appreciate these suggestions.